   

# *TrxT* and *dhd* are dispensable for Drosophila brain development but essential for *l(3)mbt* brain tumour growth

Cristina Molnar [iD][1], Jan Peter Heinen [iD][1], Jose Reina [iD][1], Salud Llamazares[1], Emilio Palumbo[2,3], Giulia Pollarolo[1,5] & Cayetano Gonzalez [iD][1,4 ✉]

## Abstract

Expression of the Drosophila cancer-germline (CG), X-linked, head-to-head gene pair *TrxT* and *dhd* is normally germline-specific but becomes upregulated in brain tumours caused by mutation in *l(3)mbt*. Here, we show that *TrxT* and *dhd* play a major synergistic role in the emergence of *l(3)mbt* tumour-linked transcriptomic signatures and tumour development, which is remarkable, taking into account that these two genes are never expressed together under normal conditions. We also show that *TrxT*, but not *dhd*, is crucial for the growth of *l(3)mbt* allografts, hence suggesting that the initial stages of tumour development and long-term tumour growth may depend on different molecular pathways. In humans, head-to-head inverted gene pairs are abundant among CG genes that map to the X chromosome. Our results identify a first example of an X-linked, head-to-head CG gene pair in Drosophila, underpinning the potential of such CG genes, dispensable for normal development and homoeostasis of somatic tissue, as targets to curtail malignant growth with minimal impact on overall health.

**Keywords** Malignant Growth; Tumours; Cancer-Testis Genes; Cancer-Germline Genes
**Subject Categories** Cancer; Development; Genetics, Gene Therapy & Genetic Disease

## Introduction

Drosophila tumour suppressor gene *l(3)mbt* encodes a conserved transcriptional regulator with ubiquitous expression (Gateff et al, 1993). Mutations affecting *l(3)mbt* result in malignant brain tumours (henceforth referred to as mbt tumours) that originate in the neuroepithelium (NE) from where they expand invading other areas of the larval brain lobe. Upon allografting into healthy adult flies, mbt tumours grow massively, invading the host's organs and ultimately leading to host mortality (Gateff et al, 1993). Remarkably, these malignant traits are much more prominent in males than in females (Molnar et al, 2019).

Gene expression profiling of mbt tumours has revealed two types of tumour-linked transcriptomic signatures: the so-called mbt tumour signature (MBTS) and the sex-dimorphic signature (SDS). The MBTS includes genes that are dysregulated in mbt tumours compared to normal brains (Janic et al, 2010). The sex-dimorphic signature (SDS) includes genes that are dysregulated in mbt tumours of one sex compared to mbt tumours of the other (Molnar et al, 2019). Both the MBTS and SDS signatures are enriched in genes whose expression in wild-type animals is restricted to the germline, hence similar to the so-called "cancer-testis" (CT) genes reported in human cancer (Janic et al, 2010; Molnar et al, 2019). We refer to these as "cancer-germline" (CG) to acknowledge the fact that some of these genes are also expressed in the female germline. Some CG genes upregulated in *l(3)mbt* individuals have been shown to play crucial roles in mbt tumour growth (Janic et al, 2010; Rossi et al, 2017).

Included in the MBTS are the thioredoxins *dhd* and *TrxT*, the latter of which is also a member of the SDS (Molnar et al, 2019). Thioredoxins are ubiquitously present in all living organisms and cellular compartments and play important roles in development and disease. Their general role is to control the cellular redox state by catalysing the reduction of oxidised substrates (Holmgren, 1985; Nishinaka et al, 2001; Powis and Montfort, 2001). However, thioredoxins also have a number of specialised functions like regulating cell death, transcription factor activity, and protein folding (Collet and Messens, 2010; Powis and Montfort, 2001; Wilkinson and Gilbert, 2004).

According to Flybase (Gramates et al, 2022), the Drosophila thioredoxin group includes 9 proteins out of which Dhd, TrxT, Trx-2 and Txl contain the classical thioredoxin motif WCGPCK (AlOkda and Van Raamsdonk, 2023; Mondal and Singh, 2022; Svensson and Larsson, 2007). Trx-2, which is ubiquitously expressed, is the orthologue of human TXN that is found at elevated levels in many cancer types and plays an important role in

[1]Institute for Research in Biomedicine (IRB Barcelona), The Barcelona Institute of Science and Technology, Carrer Baldiri Reixac, 10, 08028 Barcelona, Spain. [2]Centre for Genomic Regulation (CRG), The Barcelona Institute of Science and Technology, 08003 Barcelona, Spain. [3]Universitat Pompeu Fabra (UPF), 08002 Barcelona, Spain. [4]Institucio Catalana de Recerca i Estudis Avançats (ICREA), Pg Lluis Companys 23, 08010 Barcelona, Spain. [5]Present address: ISGlobal, Carrer del Dr. Aiguader, 88, 08003 Barcelona, Spain. ✉E-mail: gonzalez@irbbarcelona.org

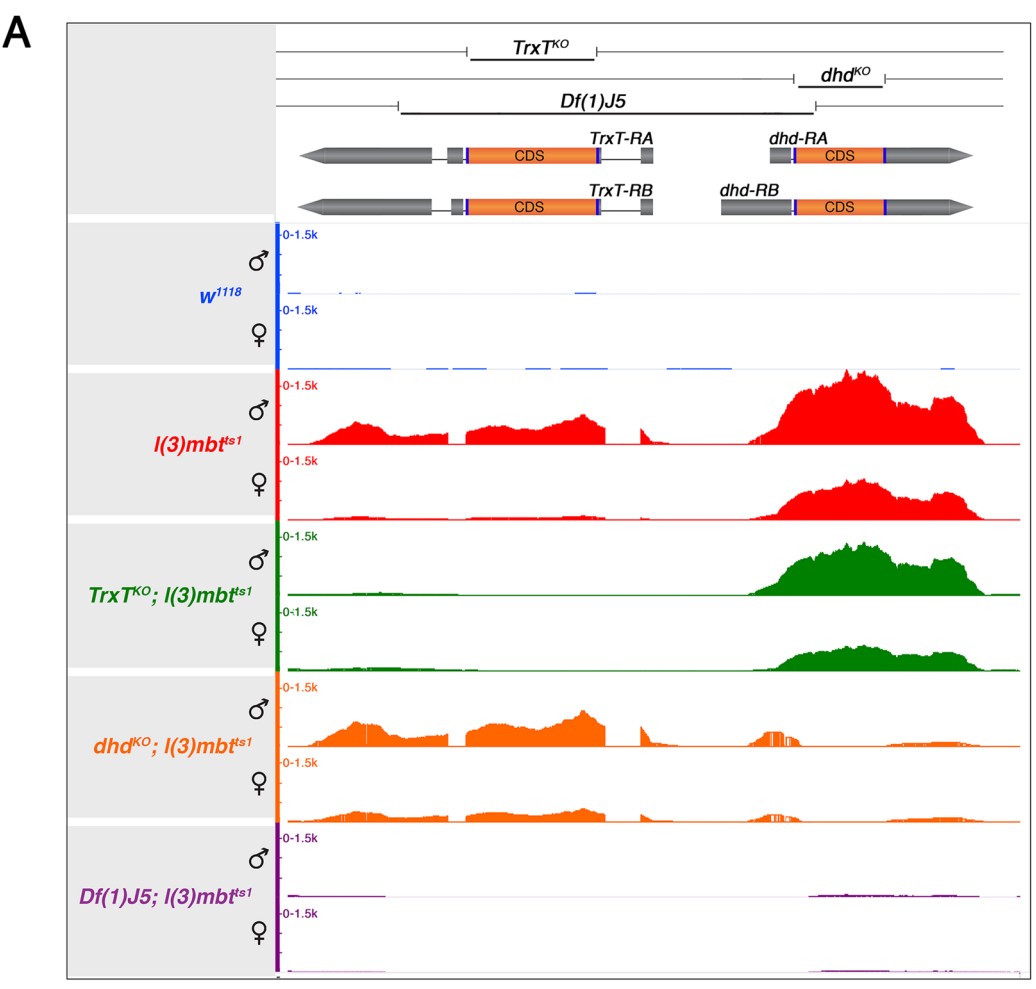

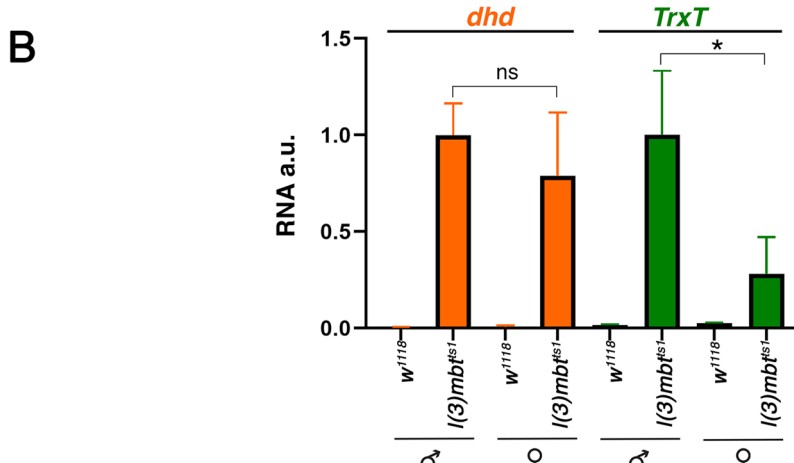

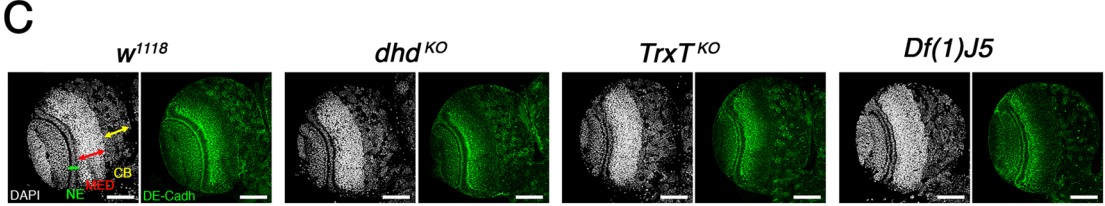

**Figure 1.** ***TrxT*** **and** ***dhd*** **are upregulated in mbt tumours and are dispensable for normal brain development.**

(A) Scheme of the *TrxT* and *dhd* genomic region showing the span of CRISPR knock-out alleles *TrxT^KO* and *dhd^KO* and *Df(1)J5*. Genome browser view of RNA-seq reads aligned to the *TrxT* - *dhd* locus in *w^TII8*, *l(3)mbt^ts1*, *TrxT^KO*; *l(3)mbt^ts1*, *dhd^KO*; *l(3)mbt^ts1*, and *Df(1)J5*; *l(3)mbt^ts1* larval brains. (B) Quantification by RT-qPCR of the *TrxT* and *dhd* transcripts in *l(3)mbt^ts1* and *w^TII8* larval brains. Error bars indicate SD of data derived from biological triplicates (two technical duplicates each). au., arbitrary units. Student's *t* test; *P < 0.05; ns P > 0.05. (C) Larval brain lobes from wild-type (*w^TII8*), *dhd^KO*, *TrxT^KO*, and *Df(1)J5* mutant larvae stained with DAPI (grey) and anti-DE-cadherin (green). Neuroepithelium (NE), medulla (MED), and central brain (CB) are labelled by green, red, and yellow arrows, respectively. None of these landmarks are affected in *dhd^KO*, *TrxT^KO*, and *Df(1)J5* larvae. Data Information: (B) statistical analysis was performed using Student's *t* test. (C) Scale bar, 50 μm. Source data are available online for this figure.

tumour progression and metastasis (Arner and Holmgren, 2006; Flores et al, 2012; Karlenius and Tonissen, 2010).

In wild-type animals, TrxT and Dhd are male- and female-germline-specific, respectively (Svensson et al, 2003), and, therefore, their unscheduled upregulation in somatic larval brain mbt tumours qualify them as Drosophila CG genes. These two proteins are encoded by a pair of adjacent genes on the X chromosome that are transcribed in opposite directions (head-to-head). This is interesting because head-to-head inverted repeats are abundant among X-linked CT genes (CT-X), which account for nearly half of all known CTs in humans (Fratta et al, 2011; Ross et al, 2005; Salmaninejad et al, 2016; Simpson et al, 2005; Stevenson et al, 2007; Warburton et al, 2004; Zhao et al, 2012).

Dhd is expressed during oogenesis and unlike TrxT and other thioredoxins, has positively charged sites on its surface, a feature that has been suggested to be linked to its role in the replacement of protamines by histones that drives sperm nuclear decondensation upon fertilisation (Emelyanov and Fyodorov, 2016; Freier et al, 2021; Tirmarche et al, 2016). In addition, Dhd plays a crucial role in the major global change in redox state that takes place at the oocyte-to-embryo transition, reducing and modulating the activity of RNA-binding proteins and histone modifiers like demethylase NO66 (Petrova et al, 2018). The majority of fertilised eggs derived from *dhd* mutant females fail to complete meiosis, and a small number of escaper embryos present defects in head formation (Salz et al, 1994).

The testis-specific TrxT has been much less studied, and no function has yet been assigned to this gene. TrxT has been reported to localise at Y chromosome loops in primary spermatocytes (Svensson et al, 2003), but the possible functional significance of this association remains unknown. Males carrying a deficiency that uncovers TrxT remain fully viable and fertile, and do not present any visible phenotype (Svensson et al, 2003). TrxT contains a highly flexible C-terminal tail that appears to stabilise the protein in a closed conformation and is not present in other canonical thioredoxins (Freier et al, 2021).

In addition to being ectopically expressed in mbt tumours, *TrxT* and *dhd* have been previously identified as suppressors of mbt growth in a high-content RNAi screen that was carried out using unsexed larval brain samples (Rossi et al, 2017). Here, using genetic loss-of-function conditions, including CRISPR knock-out alleles that we have generated to this end, and taking into account the highly sexually dimorphic nature of mbt tumours (Molnar et al, 2019), we show that *TrxT* and *dhd* exert a synergistic contribution to the development of mbt tumours that is much more substantial in male than in female larval brains. In contrast, the massive growth of allografted mbt tumours derived from male larvae is significantly dependent upon TrxT, but no Dhd function. We have also found

that the combined action of *TrxT* and *dhd* accounts for the transcriptional dysregulation of a significant fraction of the genes that compose the MBTS and most of the SDS mbt tumour signatures.

# Results and discussion

## *TrxT* and *dhd* are dispensable, alone or in combination, for normal brain development

As a first step to investigate the individual contribution of *TrxT* and *dhd* to mbt tumour growth, we generated the CRISPR knock-out alleles, *TrxT^KO* and *dhd^KO* that lack the entire CDS of each of these genes (Fig. 1A). To assess the combined effect of *TrxT* and *dhd* loss, we have made use of *Df(1)J5*, a 1.4 kb deficiency that uncovers most of the *TrxT* gene, and the entire 5'UTR plus a small part of the CDS of *dhd* (Salz et al, 1994; Svensson et al, 2003; Tirmarche et al, 2016). Consistent with the reported phenotypes of *Df(1)J5*, both *TrxT^KO* hemizygous males and *dhd^KO* homozygous females are viable and present no observable cuticular phenotypes; homozygous *dhd^KO* females are sterile; and homozygous *TrxT^KO* females, *dhd^KO* males, and *TrxT^KO* males are fully fertile (Salz et al, 1994; Svensson et al, 2003; Tirmarche et al, 2016). RNA-seq data confirms the total loss of expression of the corresponding transcripts in *TrxT^KO*; *l(3)mbt^ts1*, *dhd^KO*; *l(3)mbt^ts1*, and *Df(1)J5*; *l(3)mbt^ts1* larval brains. (Fig. 1A). RNA-seq data also shows that *TrxT* is significantly upregulated in *l(3)mbt^ts1* males compared to females (FC = 7.06; FDR = 1.10E-44) while *dhd* is not (FC = 1.89; FDR = 2.00E-14). These data are further substantiated by reverse transcription-quantitative polymerase chain reaction (RT-qPCR) (Fig. 1B).

The main histological landmarks that characterise mbt tumours, i.e.,: lamina, NE, medulla (MED), and central brain (CB), are readily visible by staining larval brain lobes with 4′,6-diamidino-2-phenylindole (DAPI) to label DNA and anti-DE-cadherin antibodies to mark the adherens junctions of the NE (Fig. 1C). Consistently with their germline-restricted expression, and the reported lack of somatic phenotype (Pellicena-Palle et al, 1997; Salz et al, 1994; Svensson et al, 2003; Tirmarche et al, 2016; Torres-Campana et al, 2022), none of these brain histological landmarks are affected in *dhd^KO*, *TrxT^KO*, and *Df(1)J5* larvae (Fig. 1C).

## The combined action of TrxT and Dhd contribute to mbt tumour development, but only TrxT plays a significant role in long-term, sustained tumour growth

Mbt tumours from male and female larvae can easily be told apart from each other on the basis of how they affect normal brain

## A

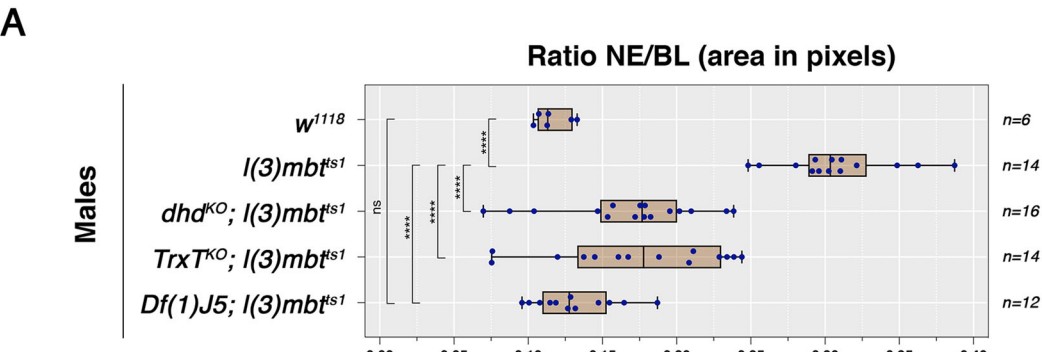

**Ratio NE/BL (area in pixels)**

Males

- $w^{1118}$ n=6
- $l(3)mbt^{ts1}$ n=14
- $dhd^{KO}; l(3)mbt^{ts1}$ n=16
- $TrxT^{KO}; l(3)mbt^{ts1}$ n=14
- $Df(1)J5; l(3)mbt^{ts1}$ n=12

## B

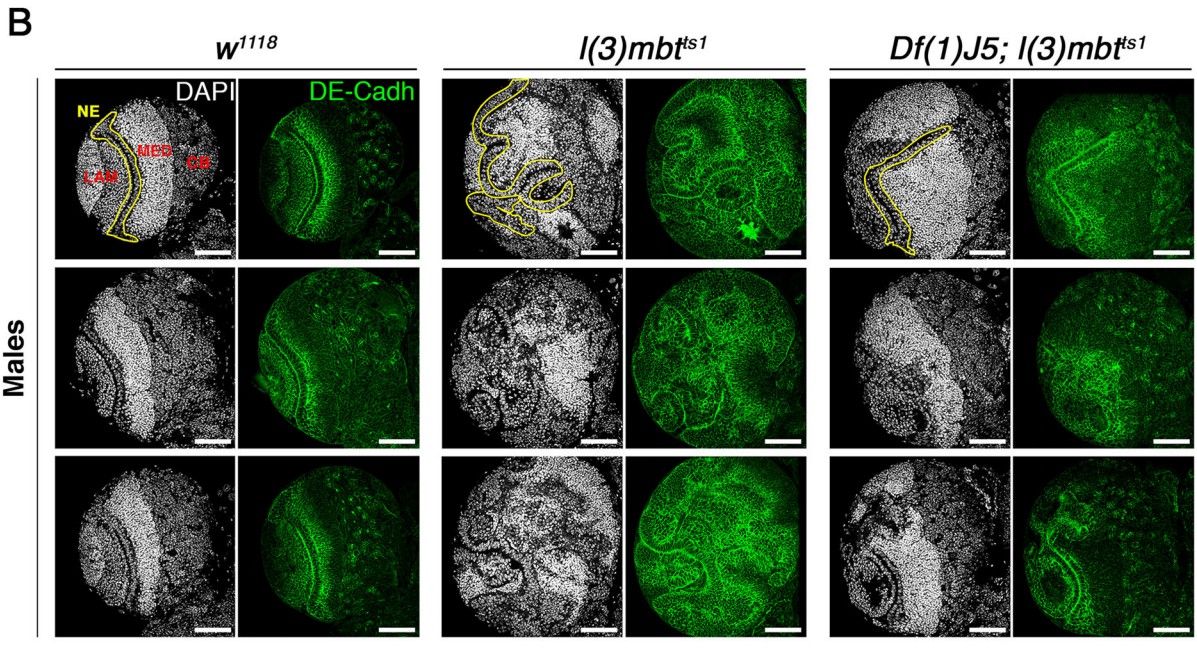

## C

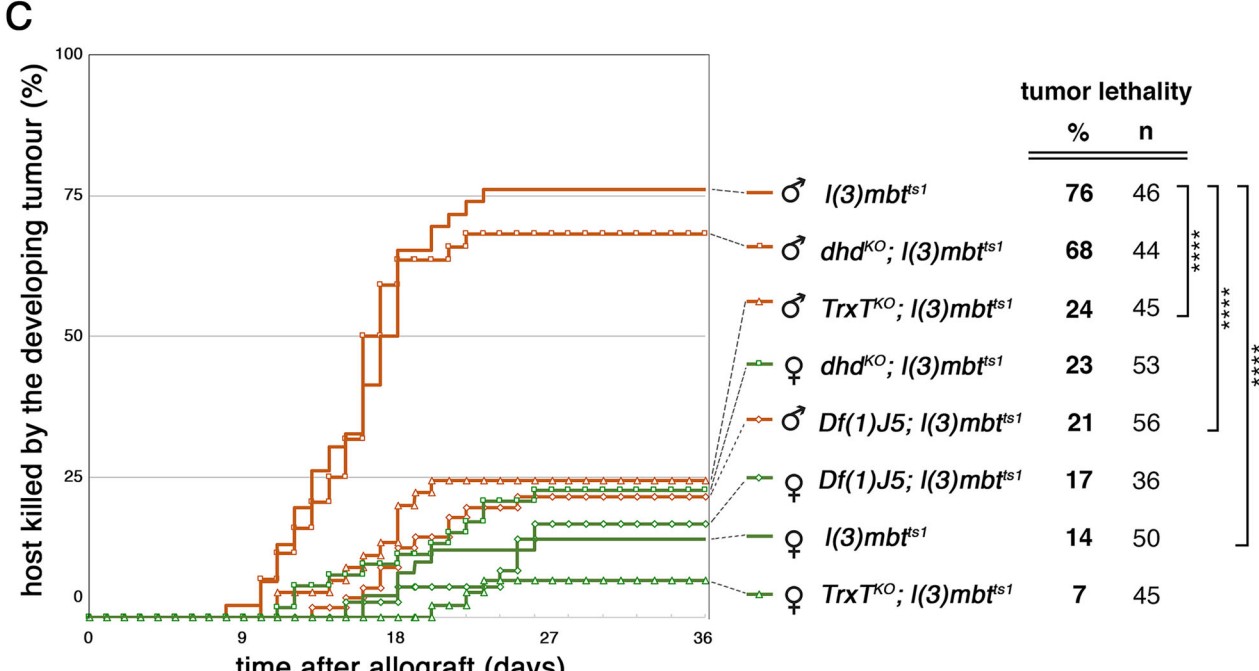

| | tumor lethality | |
|---|---|---|
| | **%** | **n** |
| ♂ $l(3)mbt^{ts1}$ | **76** | 46 |
| ♂ $dhd^{KO}; l(3)mbt^{ts1}$ | **68** | 44 |
| ♂ $TrxT^{KO}; l(3)mbt^{ts1}$ | **24** | 45 |
| ♀ $dhd^{KO}; l(3)mbt^{ts1}$ | **23** | 53 |
| ♂ $Df(1)J5; l(3)mbt^{ts1}$ | **21** | 56 |
| ♀ $Df(1)J5; l(3)mbt^{ts1}$ | **17** | 36 |
| ♀ $l(3)mbt^{ts1}$ | **14** | 50 |
| ♀ $TrxT^{KO}; l(3)mbt^{ts1}$ | **7** | 45 |

**Figure 2.** *TrxT* and *dhd* depletion inhibits male-specific mbt tumour traits.

(A) The relative size of the neuroepithelium (NE) as a fraction of brain lobe (BL) area in male control ($w^{1118}$), *l(3)mbt^{ts1}* single-mutant, and *dhd^{KO}; l(3)mbt^{ts1}*, *TrxT^{KO}; l(3)mbt^{ts1}* *Df(1)J5; l(3)mbt^{ts1}* double-mutant male larvae. ****$P < 0.0001$; ns $P > 0.05$. (B) Larval brain lobes from control ($w^{1118}$), *l(3)mbt^{ts1}* single-mutant, and *Df(1)J5; l(3)mbt^{ts1}* double-mutant male larvae stained with DAPI (grey) and anti-DE-cadherin (green). Male *l(3)mbt^{ts1}* lobes present an enlarged and convoluted NE (outlined in yellow) that expands over most of the optic lobe such that the MED is dispersed and the CB is hardly recognisable. Male *Df(1)J5; l(3)mbt^{ts1}* lobes present a notable recovery of wild-type traits. For each genotype, three representative different lobes are shown. Scale bar, 50 μm. (C) Tumour growth rate and host lethality caused by allografted *l(3)mbt^{ts1}* single-mutant, and *dhd^{KO}; l(3)mbt^{ts1}*, *TrxT^{KO}; l(3)mbt^{ts1}* and *Df(1)J5; l(3)mbt^{ts1}* double-mutants, male (brown) and female (green) brain lobes. Male *TrxT^{KO}; l(3)mbt^{ts1}* and male *Df(1)J5; l(3)mbt^{ts1}* implants develop and kill hosts at a much lower rate than *l(3)mbt^{ts1}* implants. Chi-squared test, ****$P < 0.0001$. Data Information: (A) The vertical line within each box represents the median, and the box boundaries are defined by the 25th and 75th percentiles. The whiskers extend to the minimum and maximum values. Each blue dot represents a NE/BL ratio value (area in pixels) for a single brain lobe, with sample sizes indicated ($n = 6, 14, 16, 14, 12$). (A) Statistical analysis was performed using Mann–Whitney test, or (C) Chi-squared test. (B) Scale bar, 50 μm. Source data are available online for this figure.

development (Molnar et al, 2019). Most mbt tumours derived from female larvae present shortened NE but retain a relatively normal CB and compact MED, while most male mbt tumours present an enlarged and convoluted NE that expands over most of the optic lobe such that the MED is dispersed and the CB is hardly recognisable (Figs. 2A,B and EV1).

To investigate the effect of *TrxT* and *dhd* in mbt tumour development, we analysed larval brains from *w^{1118}*, *l(3)mbt^{ts1}*, *TrxT^{KO}; l(3)mbt^{ts1}*, *dhd^{KO}; l(3)mbt^{ts1}*, and *Df(1)J5; l(3)mbt^{ts1}* larvae. We found that loss of either *TrxT*, or *dhd* results in a partial recovery of wild-type traits and that this recovery is notably enhanced by the combined loss of both thioredoxins. The effect is much more clear in males where differences between wild-type and *l(3)mbt^{ts1}* brains are prominent. Thus, for instance, in males, the size of the NE relative to the entire brain lobe (NE/BL) is highly significantly ($P = 2.5 \times 10^{-12}$) larger in *l(3)mbt^{ts1}* larval brains (NE/BL = 0.31±0.04) than in wild-type brains (NE/BL = 0.12±0.01) (Fig. 2A). The NE/BL ratio is significantly reduced -hence closer to wild-type levels-, in *TrxT^{KO}; l(3)mbt^{ts1}* and *dhd^{KO}; l(3)mbt^{ts1}* double mutants (NE/BL = 0.17±0.06 and NE/BL = 0.17±0.05, respectively), but it is even more reduced in *Df(1)J5; l(3)mbt^{ts1}* larvae (NE/BL = 0.13±0.03) to the extent of becoming indistinguishable from wild-type values ($P = 0.355$; Fig. 2A,B). The same applies to the medulla/BL and CB/BL ratios ($P = 0.8916$ and $P = 0.6165$, respectively). These effects make *Df(1)J5; l(3)mbt^{ts1}* anatomy reminiscent of that of wild-type brains (Fig. 2B). The same can be said for *Df(1)J5; l(3)mbt^{ts1}* female larvae, although given the much less severe phenotype of female mbt tumours, the effect caused by *Df(1)J5* is quantitatively minor (Figs. EV1 and EV2). Consequently, mbt tumour sex dimorphism is strongly reduced in *Df(1)J5; l(3)mbt^{ts1}* compared to *l(3)mbt^{ts1}*. From these results we conclude that the head-to-head pair *TrxT / dhd* contribute in a cooperative fashion to the development of brain tumours caused by the loss of the *l(3)mbt* tumour suppressor.

To further investigate the effect of loss-of-function conditions for *TrxT* and *dhd* in the tumourigenic potential of mbt tumours we performed allograft tests. As published (Molnar et al, 2019), we confirmed that lethality is very significantly higher in hosts implanted with *l(3)mbt^{ts1}* tissue from male larvae (76%, $n = 46$) than in hosts implanted with *l(3)mbt^{ts1}* tissue from female larvae (14%, $n = 50$) (Fig. 2C). For allografts derived from female larvae, we found that differences in lethality rate caused by *TrxT^{KO}; l(3)mbt^{ts1}*, *dhd^{KO}; l(3)mbt^{ts1}*, *Df(1)J5; l(3)mbt^{ts1}*, and *l(3)mbt^{ts1}* tissues (7–23%) were not significant (Fig. 2C). However, for allografts derived from male larvae, the effect of *TrxT^{KO}* and *Df(1)J5* was remarkable, with lethality rates dropping from 76% to 24%

($P = 9.5 \times 10^{-7}$, $n = 45$) and 21% ($P = 3.6 \times 10^{-8}$, $n = 56$), respectively. Male *dhd^{KO}; l(3)mbt^{ts1}* allografts remain nearly as lethal as *l(3)mbt^{ts1}* allografts (Fig. 2C).

As far as *Df(1)J5* is concerned, these results are consistent with its strong suppression effect on mbt larval brain tumour growth. However, while the combined loss of *TrxT* and *dhd* may account for the effect on *Df(1)J5* on mbt larval brain tumour growth, *TrxT*, and not *dhd*, seems to account for most of the reduction in host lethality rate brought about by *Df(1)J5*. These results show that unlike for in situ development of tumour traits, *TrxT* and *dhd* do not synergistically contribute to long-term tumour growth potential, hence suggesting that the initial stages of tumour development and long-term tumour growth are distinct from each other and may depend upon different molecular pathways.

### *TrxT* and *dhd* play a major synergistic contribution to the emergence of MBTS and SDS mbt tumour signatures

The remarkable recovery of wild-type larval traits brought about by the combined loss of *TrxT* and *dhd* in *l(3)mbt^{ts1}* mutant larvae opens the question of the status of expression of the genes that are dysregulated in mbt tumours. To this end, we carried out RNA-seq to identify the transcriptomes of *Df(1)J5; l(3)mbt^{ts1}*, *dhd^{KO}; l(3)mbt^{ts1}*, *TrxT^{KO}; l(3)mbt^{ts1}*, *l(3)mbt^{ts1}* and *w^{1118}*.

We first focused on transcripts that are upregulated in male mbt tumour samples compared to male wild-type larval brains (mMBTS). We identified 836 mMBTS genes (Dataset EV1). Included in this list are the majority of the genes that were originally identified as MBTS genes using Affymetrix microarrays in non-sexed samples (Janic et al, 2010) (Dataset EV1). We then investigate the effect of loss of *TrxT*, *dhd*, or both on the mMBTS. We found that compared to male *l(3)mbt^{ts1}*, only 10.8% ($n = 91$) and the 8.7% ($n = 73$) of the mMBTS genes are downregulated in the double mutants *dhd^{KO}; l(3)mbt^{ts1}*, and *TrxT^{KO}; l(3)mbt^{ts1}*, respectively. (Figs. 3A,B and EV3). However, when mbt is suppressed by depleting both *TrxT* and *dhd* simultaneously in male *Df(1)J5; l(3)mbt^{ts1}* individuals, the percentage of mMBTS downregulated genes increases to 21.7% ($n = 182$) (Fig. 3C). This effect is much more conspicuous when analysed by hierarchical clustering that shows male *Df(1)J5; l(3)mbt^{ts1}* condition to be closer to wild-type brains while *dhd^{KO}; l(3)mbt^{ts1}*, or *TrxT^{KO}; l(3)mbt^{ts1}*, remain closer to *l(3)mbt^{ts1}* (Fig. 3D). These results strongly suggest that *TrxT* and *dhd* cooperate in controlling gene expression differences between tumour and wild-type samples.

The effect of suppressors of mbt tumour larval brain phenotype on the expression of the MBTS has previously been reported for

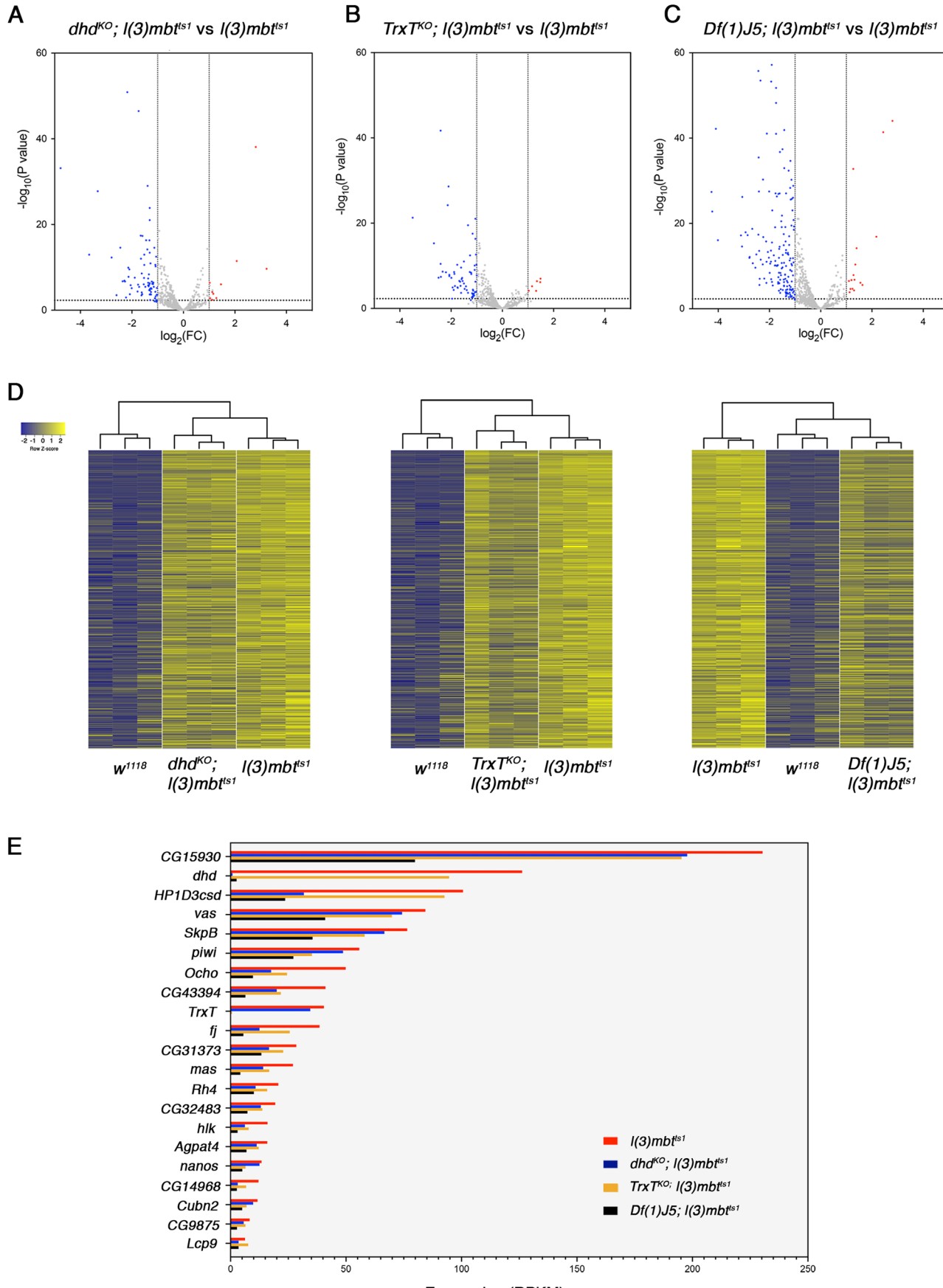

◄ **Figure 3.** ***TrxT* and *dhd* depletion reduces gene expression differences between tumour and wild-type tissues.**

(A–C) Volcano plots showing the effect of loss of *dhd* (A), *TrxT* (B), or both *TrxT* and *dhd* simultaneously (C) on the expression levels of mMBTS genes in *l(3)mbt^ts1^* mutant male brains. Red and blue dots represent genes that are significantly (FDR = 0.05) up (FC > 2) or downregulated (FC < -2), respectively. (D) Heatmaps of gene expression profiles of wild-type (*w^1118^*), mbt (*l(3)mbt^ts1^*), *dhd^KO^; l(3)mbt^ts1^*, *TrxT^KO^; l(3)mbt^ts1^* and *Df(1)J5; l(3)mbt^ts1^* male samples. Each row corresponds to one of the five hundred most significantly upregulated genes in mbt tumours compared to wild-type male samples. Expression levels are reported as Row Z-score; blue and yellow indicate low and high expression, respectively. Dendrograms on the top of the heatmaps show hierarchical clustering between samples. (E) Bar char with the set of the original MBTS genes (Janic et al, 2010) that are downregulated in *Df(1)J5; l(3)mbt^ts1^* male samples, showing their expression levels in *l(3)mbt^ts1^* (red), *dhd^KO^; l(3)mbt^ts1^* (blue), *TrxT^KO^; l(3)mbt^ts1^* (yellow), and *Df(1)J5; l(3)mbt^ts1^* (black) male samples. The genes *nanos* and *Ocho* are also significantly downregulated in *TrxT^KO^; l(3)mbt^ts1^*, and *Ocho, HP1D3csd, hlk, fj, Lcp9, CG43394*, and *CG14968* are significantly downregulated in *dhd^KO^; l(3)mbt^ts1^*. Data Information: (A–C) Differential expression analysis was performed using *edgeR*, which applies the generalised linear model (GLM) likelihood ratio statistical test. Genes with absolute FC > 2 and FDR < 0.05 were considered differentially expressed. (D) Heatmaps were performed using the function *heatmap.2* in R. (E) Expression levels of transcripts correspond to RPKM values. Source data are available online for this figure.

*Nipped-A*, *Translationally Controlled Tumour Protein* (*Tctp*), *meiotic W68* (*mei-W68*), and *PHD finger protein 7* (*Phf7*) (Molnar et al, 2019; Rossi et al, 2017). Rather counterintuitively, phenotypically wild-type like *Nipped-A^RNAi^; l(3)mbt^RNAi^*, *Tctp^RNAi^; l(3)mbt^RNAi^*, *mei-W68^1^; l(3)mbt^ts1^*, and *Phf7^ΔN2^; l(3)mbt^ts1^* double-mutant larval brains still expressed most of the MBTS to the extent that hierarchical clustering showed these four conditions to be closer to *mbt* tumours than to wild-type brains (Molnar et al, 2019; Rossi et al, 2017). Here, we have found that the same goes for *TrxT^KO^; l(3)mbt^ts1^* and *dhd^KO^; l(3)mbt^ts1^*. In contrast, concomitant removal of *TrxT* and *dhd* modifies the MBTS enough to render the transcription levels of MBTS genes in *Df(1)J5; l(3)mbt^ts1^* individuals closer to those of wild-type brains.

Downregulated mMBTS genes in *Df(1)J5; l(3)mbt^ts1^* versus *l(3)mbt^ts1^* include a quarter (21 out of 86) of the original MBTS genes (Janic et al, 2010) (Fig. 3E). Notably, four of these (*nanos, vas, piwi* and *CG15930*) are germline genes that have been shown to be required for mbt tumour growth (Janic et al, 2010; Rossi et al, 2017). Expression of *nanos* is also significantly downregulated upon *TrxT* loss, but remains unaffected by loss of *dhd*, hence suggesting that *nanos* upregulation could be critical for sustained mbt tumour growth upon allograft which is notably reduced in *TrxT^KO^; l(3)mbt^ts1^* and *Df(1)J5; l(3)mbt^ts1^*, but unaffected in *dhd^KO^; l(3)mbt^ts1^* tissue. The *vas, piwi* and *CG15930* transcripts are not significantly downregulated following either *TrxT* or *dhd* depletion alone.

We next focused on transcriptomic differences between male and female mbt tumours. As previously described (Molnar et al, 2019), plotting the expression levels of mbt tumour transcripts in male and female samples reveals two distinct clouds corresponding to the genes that are upregulated in male versus female and vice versa (Fig. 4A). We refer to these as the male and female transcriptome SDSs (Fig. 4A; M-tSDS, brown and F-tSDS, green, respectively; and Fig. EV4). We found the M-tSDS and F-tSDS clouds remain distinct but are much closer to each other in *dhd^KO^; l(3)mbt^ts1^* and *TrxT^KO^; l(3)mbt^ts1^* than in *l(3)mbt^ts1^* (Fig. EV4), but become mixed and nearly fully overlapping in *Df(1)J5; l(3)mbt^ts1^* where 97% of M-tSDS transcripts and 93% of F-tSDS transcripts do not show any significant differential expression between male and female samples (Fig. 4B). These results show that sex-biased transcription of the M-tSDS and F-tSDS genes is partially reduced in *l(3)mbt^ts1^* brains lacking either *TrxT* or *dhd*, but it is strongly suppressed upon the lack of both. *Phf7*, the only other suppressor of mbt tumour for which sex dimorphism has been studied in detail (Molnar et al, 2019), is a strong suppressor of the larval brain phenotype but is only a partial inhibitor of the SDS (Molnar et al,

2019), much less efficient than the combined loss of both thioredoxins that essentially erases all traces of transcriptional sex dimorphism in mbt tumours.

To further study the effect of *TrxT* and *dhd* depletion on the SDSs, we plotted the significance of the fold change in the expression level of M-tSDS and F-tSDS genes between *Df(1)J5; l(3)mbt^ts1^* and *l(3)mbt^ts1^* in male and female samples. We did not find any significant differences in female tissues (Fig. 4C,D). In male samples, however, we found that as much as 60% of M-tSDS and 70% F-tSDS genes are expressed at significantly lower and higher levels, respectively in *Df(1)J5; l(3)mbt^ts1^* compared to *l(3)mbt^ts1^* (Fig. 4E,F). The same trend is observed following depletion of either one of the two thioredoxins alone, although the percentage of affected genes is always smaller: 21% and 30% of the of M-tSDS and F-tSDS in the case of *dhd^KO^; l(3)mbt^ts1^* and 15.7% and 24% of the of M-tSDS and F-tSDS in the case of *TrxT^KO^; l(3)mbt^ts1^* (Fig. EV4). Downregulated M-tSDS genes in male mbt tumours suppressed by depletion of both *TrxT* and *dhd* include *HP1D3csd, piwi, nanos, Cubn2, CG15930* and *CG14968*, which also belong to the MBTS and are known to be required for mbt tumour growth (Janic et al, 2010; Rossi et al, 2017). Venn diagrams showing the overlap between the affected M-tSDS and F-tSDS genes in the *TrxT^KO^; l(3)mbt^ts1^*, *dhd^KO^; l(3)mbt^ts1^*, and *Df(1)J5; l(3)mbt^ts1^* conditions, and GO analysis of or overlapping and non-overlapping genes are shown in Fig. EV5. From these data we conclude that *TrxT* and *dhd* play a strong synergistic effect in sex-linked gene expression dysregulation in male mbt tumour tissue by up-regulating M-tSDS genes and downregulating F-tSDS genes.

Altogether, our results identify the combined action of TrxT and Dhd as an upstream function required for the expression of the MBTS and SDS tumour signatures and for the development of the enhanced malignancy traits presented by mbt tumours in male individuals. This critical tumorigenic interaction between *TrxT* and *dhd* is quite remarkable taking into account that, under normal conditions, these two genes are never expressed together.

In humans, CT/CG genes have been identified in a wide range of cancer types (Hofmann et al, 2008). Notably, nearly half of known human CT/CG genes map to the X chromosome (a.k.a.CT-X genes) (Ross et al, 2005; Stevenson et al, 2007) and unlike CT genes encoded in autosomes, most CT-X genes belong to gene families that are organised into complex tandem, direct, or inverted (head-to-head) repeats (Fratta et al, 2011; Ross et al, 2005; Salmaninejad et al, 2016; Simpson et al, 2005; Warburton et al, 2004; Zhao et al, 2012). Notably, some head-to-head CT-X gene pairs work as a unit in human cancer (Barger et al, 2021).

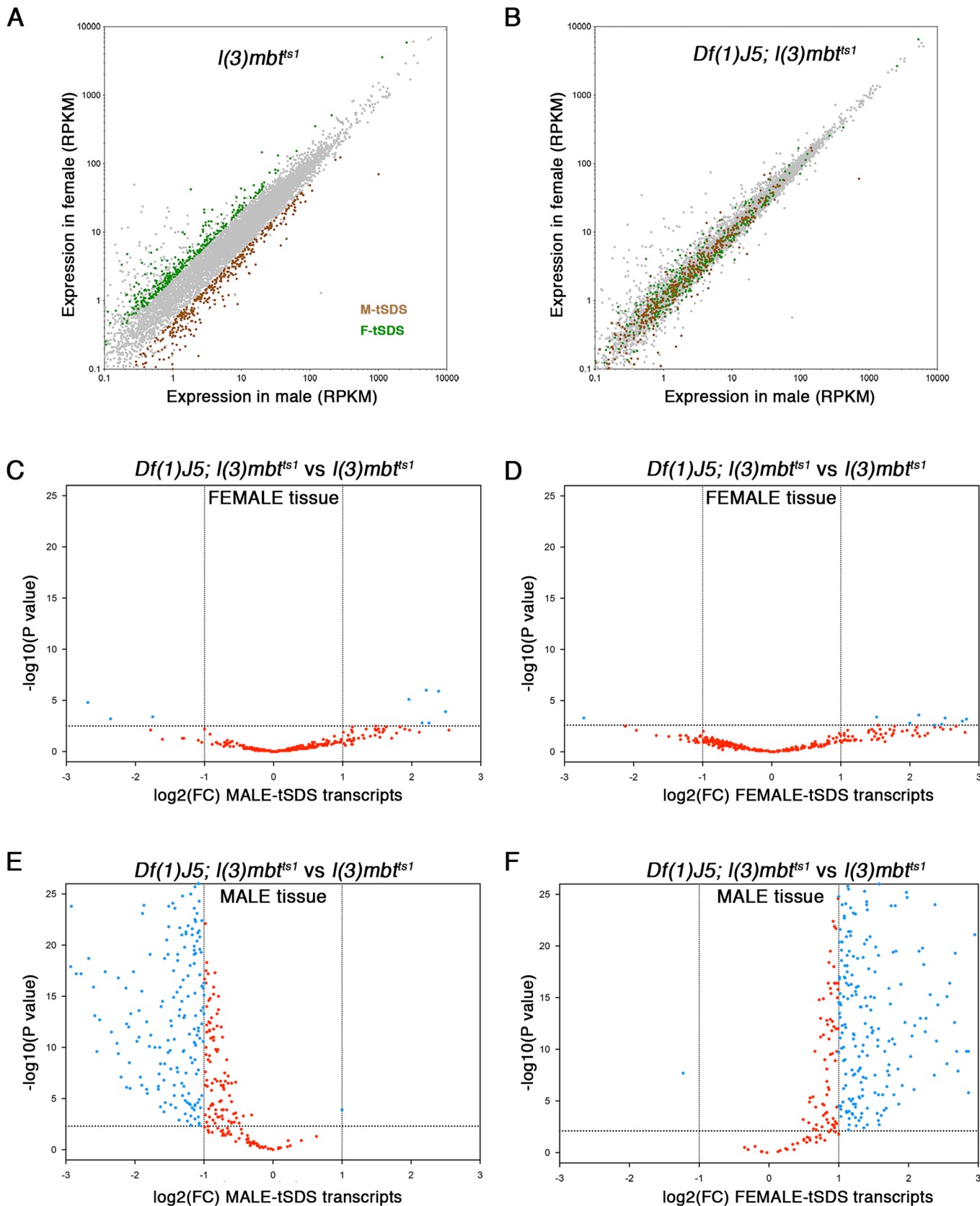

◄

**Figure 4. *TrxT* and *dhd* depletion completely suppress the sex-biased expression of mbt sex-linked dimorphic signatures.**

(A, B) Plots showing the expression level of transcripts in male (*x* axis) and female (*y* axis) samples from *l(3)mbt^ts1^* (A) and *Df(1)J5; l(3)mbt^ts1^* (B) larvae. Green and brown dots correspond to genes that are significantly overexpressed in male versus female (M-tSDS) and female versus male (F-tSDS) mbt tumours, respectively. Grey dots correspond to genes that are expressed at levels that are not significantly different between males and females. (C–F) Volcano plots showing the significance of the fold change in expression levels of the M-tSDS (C, E) and F-tSDS (D, F) genes between *Df(1)J5; l(3)mbt^ts1^* and *l(3)mbt^ts1^* in female (C, D) and male (E, F) samples. Blue dots represent genes that are significantly (FDR = 0.05) upregulated (log2(FC) > 1) or downregulated (log2(FC < -1) in *Df(1)J5; l(3)mbt^ts1^* compared to *l(3)mbt^ts1^*. Genes whose differential expression is not significant are coloured in red. Data Information: (A, B) Expression levels of transcripts correspond to RPKM values. (C–F) Differential expression analysis was performed using *edgeR*, which applies the generalised linear model (GLM) likelihood ratio statistical test. Genes with absolute FC > 2 and FDR < 0.05 were considered differentially expressed. Source data are available online for this figure.

Our results reveal the first instance of a X-linked, head-to-head CG pair in Drosophila and underpin the potential of cancer-germline genes that are dispensable for somatic cell development as targets to curtail malignant growth. This is particularly so for genes like *TrxT* that play no major role in development but have a substantial contribution to mbt malignant growth.

# Methods

## Fly stocks

The following mutant alleles were used in this study: *l(3)mbt^ts1^* (Gateff et al, 1993), *Df(1)J5* (Salz et al, 1994), *TrxT^KO^* and *dhd^KO^* (both generated in this work). The wild-type strain used was *w^1118^*. To distinguish male from female mbt mutant larvae, the strain *Dp(1:Y)y+* was used. Because of the temperature-sensitive condition of mbt, all crosses, including controls, were maintained at 29 °C. Double mutants *TrxT^KO^; l(3)mbt^ts1^, dhd^KO^; l(3)mbt^ts1^* and *Df(1)J5; l(3)mbt^ts1^* were generated using standard genetic techniques.

## CRISPR/Cas9-mediated knock-out alleles

*TrxT^KO^* and *dhd^KO^* mutants were made by gene editing using CRISPR-Cas9 and homologous direct repair in InDroso, France. Briefly, *nos-Cas9* embryos were injected with a vector containing a reporter cassette with an *attP–Lox– 3xP3-dsRED–Lox* flanked by the homologue donor arms and two gRNA-expressing vectors. This resulted in the full replacement of the CDS of *TrxT* and *dhd* by the reporter cassette. Transgenic lines were selected visually for the dsRED marker. By PCR and sequencing, the correct replacement of the CDSs was confirmed. The gRNAs used were the following: dhd1-AATTAAA-GAACTTGACGTGATGG, dhd2-TACATAAACGAACACAAAT-GAGG, TrxT1-AAGTCATGGTGTACCCAGTGCGG and TrxT2-CCACGACGAGAATGCCGTTCTGG.

## Genotypes and crossing schemes

Crosses for the mbt condition were done with females *yw; l(3)mbt^ts1^/TM6B,Tb* and males *yw/Dp(1:Y)y + ; l(3)mbt^ts1^/TM6B,Tb*. The progeny used were males *yw/Dp(1:Y)y + ; l(3)mbt^ts1^* and females *yw; l(3)mbt^ts1^*.

For the double-mutant *TrxT^KO^; l(3)mbt^ts1^* condition, females *yw TrxT^KO^; l(3)mbt^ts1^/TM6B,Tb* were crossed with males *yw TrxT^KO^/Dp(1:Y)y + ; l(3)mbt^ts1^/TM6B,Tb*. The progeny used were males *yw TrxT^KO^/Dp(1:Y)y + ; l(3)mbt^ts1^* and females *yw TrxT^KO^; l(3)mbt^ts1^*.

For the double-mutant *dhd^KO^; l(3)mbt^ts1^* condition, females *yw dhd^KO^/FM7,P{ActGFP}; l(3)mbt^ts1^/TM6B,Tb* were crossed with males *yw dhd^KO^/Dp(1:Y)y + ; l(3)mbt^ts1^/TM6B,Tb*. The progeny used were males *yw dhd^KO^/Dp(1:Y)y + ; l(3)mbt^ts1^* and females *yw dhd^KO^; l(3)mbt^ts1^*.

For the double-mutant *Df(1)J5; l(3)mbt^ts1^* condition, females *Df(1)J5/FM7,P{ActGFP}; l(3)mbt^ts1^/TM6B,Tb* were crossed with males *mRFP Df(1)J5/Dp(1:Y)y + ; l(3)mbt^ts1^/TM6B,Tb*. The progeny used were males *Df(1)J5/Dp(1:Y)y + ; l(3)mbt^ts1^* and females *mRFP Df(1)J5 / Df(1)J5; l(3)mbt^ts1^*.

## Immunohistochemistry

Immunostaining of whole larval brains was performed as described (Gonzalez and Glover, 1993). Briefly, brains were dissected in phosphate-buffered saline (PBS), fixed in 4% formaldehyde, rinsed in PBS-0.3% Triton X-100 (PBST), and blocked in PBST with 10% foetal calf serum (PBSTF). Primary and secondary antibodies were incubated in PBSTF overnight at 4 °C. The primary antibody used in this study was rat anti-DE-Cadherin (DCAD2, 1:100, DSHB). We used Alexa Fluor Secondary Antibodies (1:1000, Life Technologies). DNA was stained with DAPI. Larval brains were mounted in Vectashield (Molecular Probes). Images were acquired with a SP8 Leica confocal image microscope and processed in Adobe Photoshop CS6 and ImageJ.

## Quantification and statistical analysis

Eggs were collected for 24 h and allowed to develop for up to 6 days (132 + /-12h AEL) for all the experiments, except for control *w^1118^* larvae that were dissected at 5 days AEL. The ratio area of neuroepithelium/area of the brain lobe (ratio NE/BL) was calculated using images acquired with a SP8 Leica confocal image microscope and measuring the areas corresponding to the neuroepithelium and the brain lobe using ImageJ software. The results were represented in box-plots and *P* values were calculated using GraphPad Prism 10.2.0 for macOS, GraphPad Software, La Jolla California USA, www.graphpad.com. For statistical analysis, unpaired two-tailed Student's *t* test were used for normally distributed samples and equal variances, and non-parametric Mann–Whitney test were used when data were not normally distributed. Kruskal–Wallis test was used for multiple comparisons. In all cases, differences were considered significant at *P* < 0.05. All genotypes and crosses were done as described above.

## Allograft assays

Larval brain lobes grafts were carried out in female hosts as described in (Rossi and Gonzalez, 2015). Tumour lethality (%) was calculated by the number of hosts killed by the developing tumour

over the total of allografted adults. Implanted hosts were kept at 29 °C. The *P* values for the differences in tumour lethality were calculated using Chi-square test.

## Transcriptomics

### RNA-seq samples preparation and sequencing

RNA was isolated using magnetic beads (RNAClean XP, Beckman Coulter, A63987) from 10 Drosophila larval brains per sample following the protocol described in (Janic et al, 2010). RNA poly A purification, cDNA generation, adaptor ligation, library amplification and sequencing were performed as described in (Molnar et al, 2019).

### RNA-seq data processing and analysis

Data has been processed and analysed as described in (Molnar et al, 2019). Expression coverage files in BAM format (https://genome.ucsc.edu/goldenPath/help/bam.html) were generated using STAR (v2.4.0j) and later normalised according to the scaling factors obtained by the TMM normalisation method. The resulting files have been visually inspected with the Integrated Genome Browser (http://igb.bioviz.org/) software.

### Differential expression analysis

Pairwise comparisons were performed to identify differentially expressed (DE) genes between females and males and genotypes, using edgeR. Genes with fold change >2 and FDR < 0.05 were considered differentially expressed.

### Gene expression analysis by RT-qPCR

Total RNA was isolated from dissected larval brains using proteinase K and deoxyribonuclease (DNase) (Invitrogen) and purified using magnetic beads. RNA yield and quality were assessed with Qubit, followed by reverse transcription using random hexamers. Transcript levels were measured with PowerUp SYBR Green Master Mix in QuantStudio 6 Flex System (Applied Biosystems). Initial activation was performed at 95 °C for 20 s, followed by 40 cycles of 95 °C for 5 s and 60 °C for 15 s. The melting curve was generated ranging from 50 °C to 95 °C with an increment of 0.5 °C every 5 s. Primers specifics for the transcript *TrxT* and *dhd* used for RT-qPCR are as follows: TrxT-FW-AGATAATAGCACC-CAAGCTGGA, TrxT-RV-ATTCGACCGTAATGTCCTCGT dhd-FW-GATGTGGACAAATTCGAGGAGC and dhd-RV -GAGC-CAAGCGTCGATTTT. Measurements were performed on biological triplicates, with technical duplicates of each biological sample. RNA levels were normalised to *rp49*. The relative transcript levels were calculated using the $2^{-\Delta\Delta CT}$ method (Livak and Schmittgen, 2001).

### Heatmaps

Hierarchical clustering was done using the function heatmap.2 in R to generate a plot in which samples (columns) are clustered (dendogram); genes (rows) are scaled by "rows"; distance = Euclidean; and hclust method = complete linkage. Expression levels are reported as Row Z-score.

### Venn diagrams

Venn diagrams were done using the web application BioVenn (Hulsen et al, 2008).

### Gene Ontology (GO) enrichment analysis

Functional annotation of GO terms for Biological Process was performed using the online tool Database and Annotation, Visualisation and Integrated Discovery (DAVID 2021) (Sherman et al, 2022).

## Data availability

The RNA-seq raw data discussed in this publication have been deposited in NCBI's Gene Expression Omnibus and are accessible through GEO Series accession number GSE263157.

The source data of this paper are collected in the following database record: biostudies:S-SCDT-10_1038-S44319-024-00154-1.

## Peer review information

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

## Acknowledgements

The authors are very grateful to the Functional Genomics Facility of IRB Barcelona for their technical assistance; Jan Larsson and the Bloomington Stock Center for fly strains; the Developmental Studies for Hybridoma Bank for antibodies; and to all members of our laboratory for very helpful discussions. This research was supported by grants PGC2018-097372-B-100 funded by the Ministry of Science, Innovation, and Universities-Spanish State Research Agency and ERC AdG 2011 294603.

## Author contributions

**Cristina Molnar**: Data curation; Formal analysis; Investigation; Methodology; Writing—original draft; Writing—review and editing. **Jan Peter Heinen**: Investigation. **Jose Reina**: Investigation. **Salud Llamazares**: Investigation. **Emilio Palumbo**: Data curation; Formal analysis. **Giulia Pollarolo**: Data curation; Formal analysis. **Cayetano Gonzalez**: Conceptualisation; Supervision; Funding acquisition; Validation; Visualisation; Writing—original draft; Writing—review and editing.

Source data underlying figure panels in this paper may have individual authorship assigned. Where available, figure panel/source data authorship is listed in the following database record: biostudies:S-SCDT-10_1038-S44319-024-00154-1.

## Disclosure and competing interests statement

The authors declare no competing interests.

# Expanded View Figures

**A**

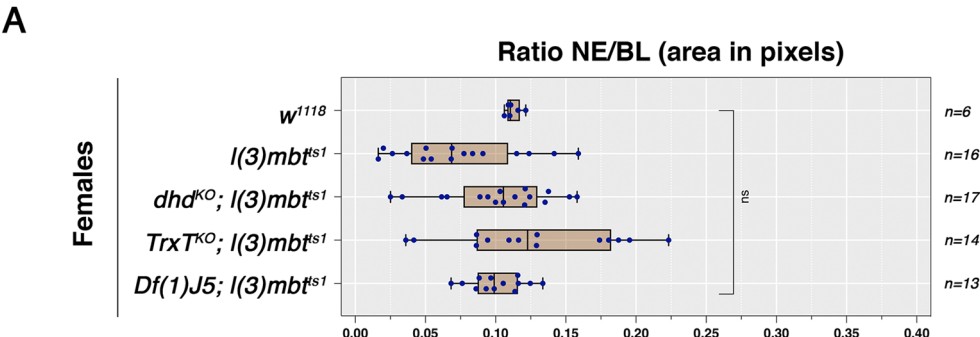

**B**

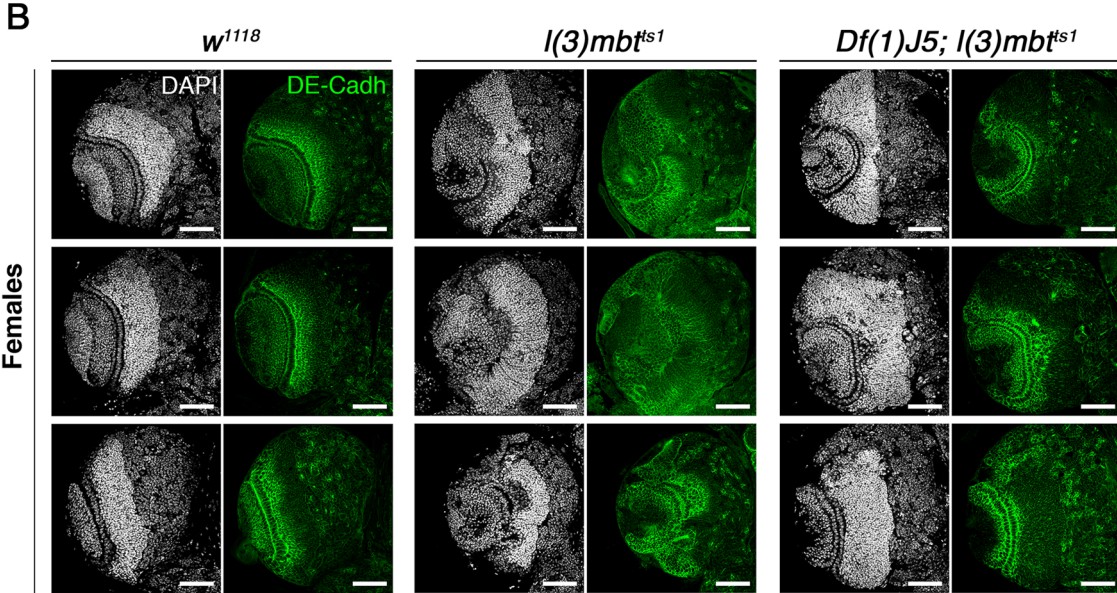

**C**

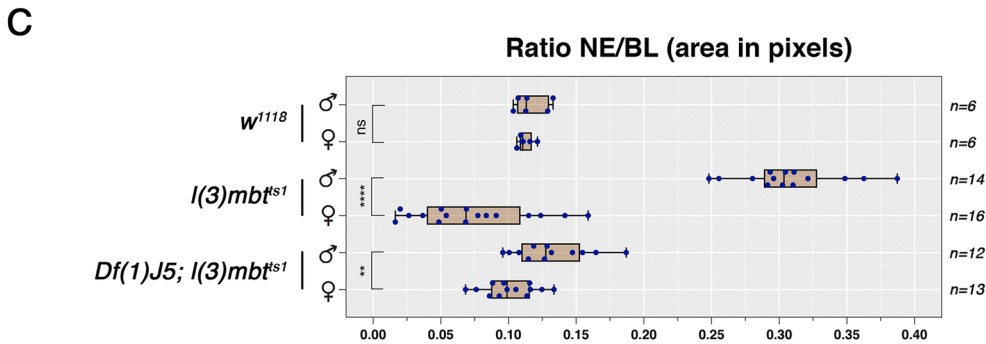

**Figure EV1.** *TrxT* and *dhd* depletion affects female-specific mbt tumour traits.

(A) Relative size of NE as a fraction of BL area in female control (*w¹¹¹⁸*), *l(3)mbtᵗˢ¹* single-mutant, and *dhdᴷᴼ; l(3)mbtᵗˢ¹*, *TrxTᴷᴼ; l(3)mbtᵗˢ¹*, *Df(1)J5; l(3)mbtᵗˢ¹* double-mutant female larvae. Kruskal–Wallis test, ns P > 0.05. (B) Larval brain lobes from control (*w¹¹¹⁸*), *l(3)mbtᵗˢ¹* single-mutant, and *Df(1)J5; l(3)mbtᵗˢ¹* double-mutant female larvae stained with DAPI (grey) and anti-DE-cadherin (green). Female *l(3)mbtᵗˢ¹* lobes present shortened NE but retain a relatively normal CB and compact medulla (MED). Female *Df(1)J5; l(3)mbtᵗˢ¹* lobes closely resemble wild-type brains. For each genotype, three representative different lobes are shown. Scale bar, 50 μm. (C) Relative size of NE as a fraction of brain lobe BL area in male and female control (*w¹¹¹⁸*), *l(3)mbtᵗˢ¹* single-mutant, and *Df(1)J5; l(3)mbtᵗˢ¹* double-mutant larvae. Student's *t* test, ****P < 0.0001; **P < 0.01; ns P > 0.05. Data Information: (A, C) The vertical line within each box represents the median, and the box boundaries are defined by the 25th and 75th percentiles. The whiskers extend to the minimum and maximum values. Each blue dot represents a NE/BL ratio value (area in pixels) for a single brain lobe, with sample sizes indicated (A: n = 6, 16, 17, 14, 13; C: n = 6, 6, 14, 16, 12, 13). (A) Statistical analysis was performed using Kruskal–Wallis test, or (C) Student's *t* test. (B) Scale bar, 50 μm.

## A

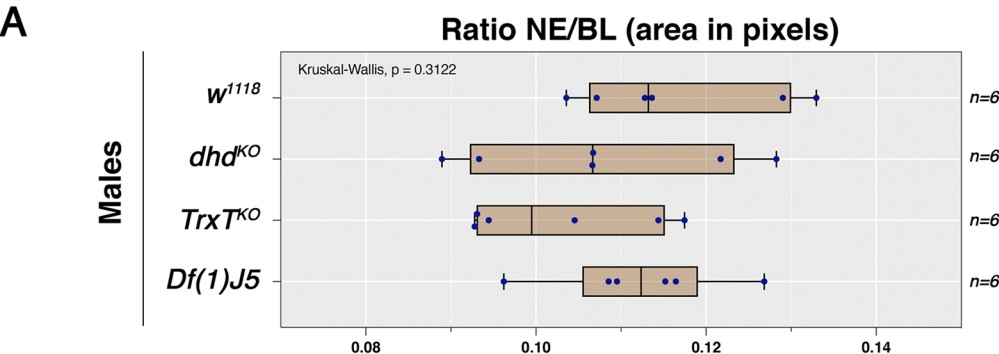

**Ratio NE/BL (area in pixels)**

Kruskal-Wallis, p = 0.3122

Males: $w^{1118}$ — n=6; $dhd^{KO}$ — n=6; $TrxT^{KO}$ — n=6; $Df(1)J5$ — n=6

## B

$dhd^{KO}$; $l(3)mbt^{ts1}$  |  $TrxT^{KO}$; $l(3)mbt^{ts1}$

DAPI   DE-Cadh

Males

Females

◀ **Figure EV2.** *TrxT* and *dhd* depletion in wild-type and mbt tumour brains.

(A) Relative size of the NE as a fraction of BL area in male control ($w^{1118}$), $dhd^{KO}$, $TrxT^{KO}$, and *Df(1)J5* male larvae. Kruskal–Wallis test, ns *P* > 0.05. (B) Larval brain lobes from $dhd^{KO}$; *l(3)mbt^{ts1}* and $TrxT^{KO}$; *l(3)mbt^{ts}* male and female larvae stained with DAPI (grey) and anti-DE-cadherin (green). Lobes corresponding to the three best mbt-suppressed phenotype are shown for each genotype. Scale bar, 50 µm. Data Information: (A) The vertical line within each box represents the median, and the box boundaries are defined by the 25th and 75th percentiles. The whiskers extend to the minimum and maximum values. Each blue dot represents a NE/BL ratio value (area in pixels) for a single brain lobe, with sample sizes indicated ($n = 6$ in all samples). (A) Statistical analysis was performed using Kruskal–Wallis test. (B) Scale bar, 50 µm.

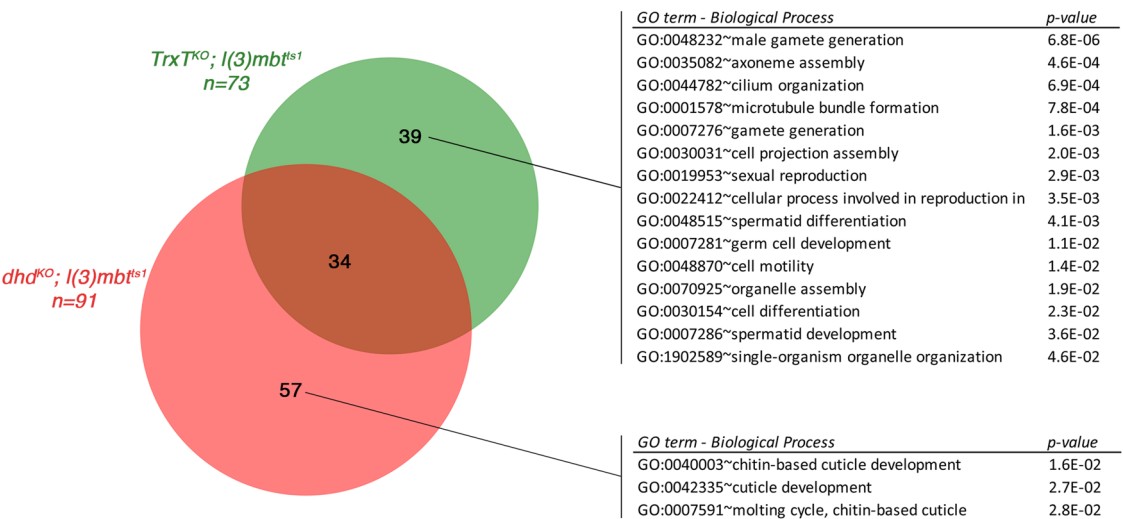

**mMBTS**

| GO term - Biological Process | p-value |
|---|---|
| GO:0048232~male gamete generation | 6.8E-06 |
| GO:0035082~axoneme assembly | 4.6E-04 |
| GO:0044782~cilium organization | 6.9E-04 |
| GO:0001578~microtubule bundle formation | 7.8E-04 |
| GO:0007276~gamete generation | 1.6E-03 |
| GO:0030031~cell projection assembly | 2.0E-03 |
| GO:0019953~sexual reproduction | 2.9E-03 |
| GO:0022412~cellular process involved in reproduction in | 3.5E-03 |
| GO:0048515~spermatid differentiation | 4.1E-03 |
| GO:0007281~germ cell development | 1.1E-02 |
| GO:0048870~cell motility | 1.4E-02 |
| GO:0070925~organelle assembly | 1.9E-02 |
| GO:0030154~cell differentiation | 2.3E-02 |
| GO:0007286~spermatid development | 3.6E-02 |
| GO:1902589~single-organism organelle organization | 4.6E-02 |

| GO term - Biological Process | p-value |
|---|---|
| GO:0040003~chitin-based cuticle development | 1.6E-02 |
| GO:0042335~cuticle development | 2.7E-02 |
| GO:0007591~molting cycle, chitin-based cuticle | 2.8E-02 |

**Figure EV3.  mMBTS genes affected by depletion of *TrxT* or *dhd* in mbt tumours.**

Venn diagram showing the number of genes of the mMBTS downregulated in males *dhd^KO; l(3)mbt^ts1* (red) and *TrxT^KO; l(3)mbt^ts1*(green) compared to *l(3)mbt^ts1*. Gene Ontology terms that are significantly enriched in the group of genes downregulated in *TrxT^KO; l(3)mbt^ts1* only, and in *dhd^KO; l(3)mbt^ts1* only ($P < 0.05$). No significantly enriched GOs were identified in the overlapping genes. Data Information: Venn Diagram was performed using BioVenn online tool, and Gene Ontology terms were analysed using DAVID 2021 online tool. Statistical analysis performed by DAVID was Fisher's Exact test.

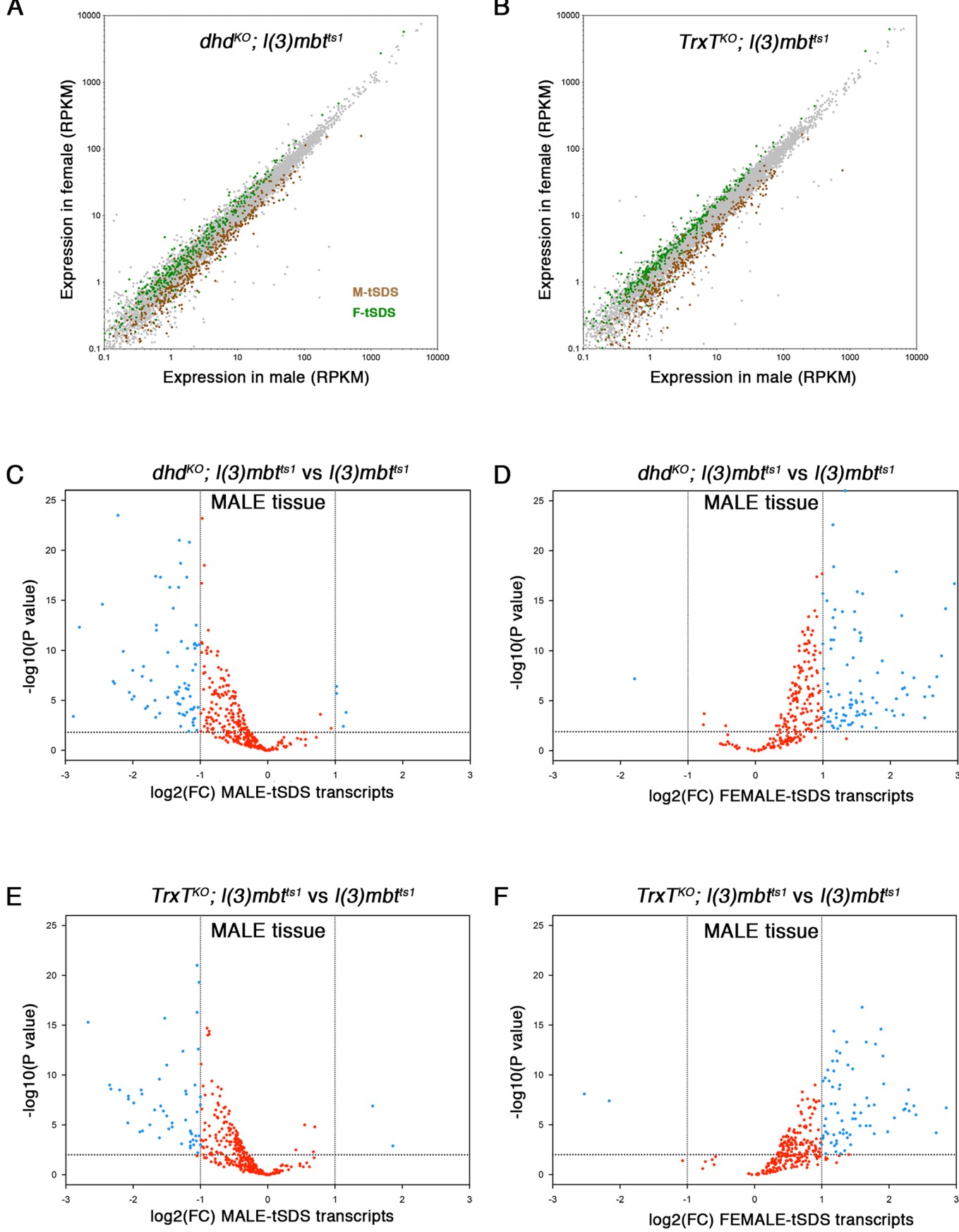

**Figure EV4. Expression of mbt SDS genes in *dhd*^KO^; *l(3)mbt*^ts1^ and *TrxT*^KO^; *l(3)mbt*^ts1^ double-mutant larvae.**

(A, B) Plots showing the expression level of transcripts in male (*x* axis) and female (*y* axis) samples from *dhd*^KO^; *l(3)mbt*^ts1^ (A) and *TrxT*^KO^; *l(3)mbt*^ts1^ (B) larvae. Green and brown dots correspond to genes that are significantly overexpressed in male versus female (M-tSDS) and female versus male (F-tSDS) mbt tumours, respectively. Grey dots correspond to genes that are expressed at levels that are not significantly different between males and females. (C–F) Volcano plots showing the significance of the fold change in expression levels of the M-tSDS (C, E) and F-tSDS (D, F) genes between *dhd*^KO^; *l(3)mbt*^ts1^ and *l(3)mbt*^ts1^ (C, D) and *TrxT*^KO^; *l(3)mbt*^ts1^ and *l(3)mbt*^ts1^ (E, F) male samples. Blue dots represent genes that are significantly (FDR = 0.05) upregulated (log2(FC) > 1) or downregulated (log2(FC < -1). Genes whose differential expression is not significant are coloured in red. Data Information: (A, B) Expression levels of transcripts correspond to RPKM values. (C–F) Differential expression analysis was performed using *edgeR*, which applies the generalised linear model (GLM) likelihood ratio statistical test. Genes with absolute FC > 2 and FDR < 0.05 were considered differentially expressed.

**A** M-tSDS down-regulated in males

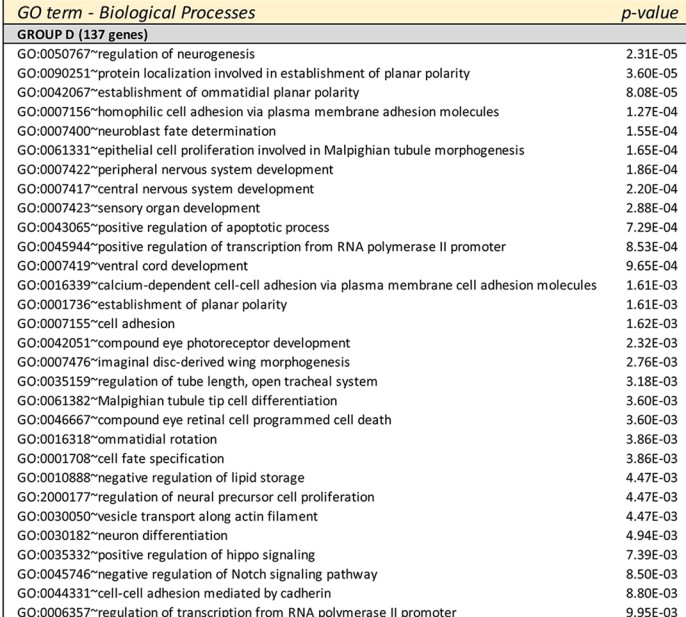

**B** F-tSDS up-regulated in males

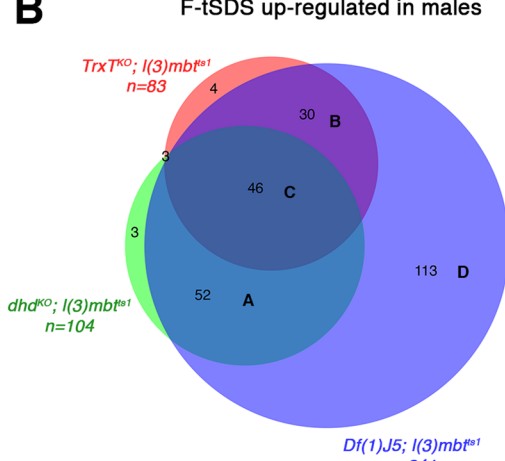

| GO term - Biological Processes | p-value |
|---|---|
| **GROUP D (137 genes)** | |
| GO:0050767~regulation of neurogenesis | 2.31E-05 |
| GO:0090251~protein localization involved in establishment of planar polarity | 3.60E-05 |
| GO:0042067~establishment of ommatidial planar polarity | 8.08E-05 |
| GO:0007156~homophilic cell adhesion via plasma membrane adhesion molecules | 1.27E-04 |
| GO:0007400~neuroblast fate determination | 1.55E-04 |
| GO:0061331~epithelial cell proliferation involved in Malpighian tubule morphogenesis | 1.65E-04 |
| GO:0007422~peripheral nervous system development | 1.86E-04 |
| GO:0007417~central nervous system development | 2.20E-04 |
| GO:0007423~sensory organ development | 2.88E-04 |
| GO:0043065~positive regulation of apoptotic process | 7.29E-04 |
| GO:0045944~positive regulation of transcription from RNA polymerase II promoter | 8.53E-04 |
| GO:0007419~ventral cord development | 9.65E-04 |
| GO:0016339~calcium-dependent cell-cell adhesion via plasma membrane cell adhesion molecules | 1.61E-03 |
| GO:0001736~establishment of planar polarity | 1.61E-03 |
| GO:0007155~cell adhesion | 1.62E-03 |
| GO:0042051~compound eye photoreceptor development | 2.32E-03 |
| GO:0007476~imaginal disc-derived wing morphogenesis | 2.76E-03 |
| GO:0035159~regulation of tube length, open tracheal system | 3.18E-03 |
| GO:0061382~Malpighian tubule tip cell differentiation | 3.60E-03 |
| GO:0046667~compound eye retinal cell programmed cell death | 3.60E-03 |
| GO:0016318~ommatidial rotation | 3.86E-03 |
| GO:0001708~cell fate specification | 3.86E-03 |
| GO:0010888~negative regulation of lipid storage | 4.47E-03 |
| GO:2000177~regulation of neural precursor cell proliferation | 4.47E-03 |
| GO:0030050~vesicle transport along actin filament | 4.47E-03 |
| GO:0030182~neuron differentiation | 4.94E-03 |
| GO:0035332~positive regulation of hippo signaling | 7.39E-03 |
| GO:0045746~negative regulation of Notch signaling pathway | 8.50E-03 |
| GO:0044331~cell-cell adhesion mediated by cadherin | 8.80E-03 |
| GO:0006357~regulation of transcription from RNA polymerase II promoter | 9.95E-03 |

| GO term - Biological Processes | p-value |
|---|---|
| **GROUP B (30 genes)** | |
| GO:0046839~phospholipid dephosphorylation | 7.23E-07 |
| GO:0006644~phospholipid metabolic process | 1.94E-05 |
| GO:0007602~phototransduction | 4.88E-03 |
| | |
| **GROUP D (113 genes)** | |
| GO:0050808~synapse organization | 6.76E-05 |
| GO:0045433~male courtship behavior, veined wing generated song product | 1.45E-04 |
| GO:0007186~G-protein coupled receptor signaling pathway | 8.31E-04 |
| GO:0002121~inter-male aggressive behavior | 6.26E-03 |
| GO:0007218~neuropeptide signaling pathway | 9.94E-03 |

**Figure EV5. mbt SDS genes affected by depletion of *TrxT* and *dhd* in mbt tumours.**

Venn diagrams and Gene Ontology terms that are significantly enriched ($P < 0.01$) in the downregulated M-tSDS (**A**) and upregulated F-tSDS (**B**) genes in *dhd*$^{KO}$; *l(3)mbt*$^{ts1}$ (green) and *TrxT*$^{KO}$; *l(3)mbt*$^{ts1}$ (red) and *Df(1)J5*; *l(3)mbt*$^{ts1}$ (blue) male larvae. A = overlapping genes between *dhd*$^{KO}$; *l(3)mbt*$^{ts1}$ and *Df(1)J5*; *l(3)mbt*$^{ts1}$; B = overlapping genes between *TrxT*$^{KO}$; *l(3)mbt*$^{ts1}$ and *Df(1)J5*; *l(3)mbt*$^{ts1}$; C = overlapping genes between *dhd*$^{KO}$; *l(3)mbt*$^{ts1}$, *TrxT*$^{KO}$; *l(3)mbt*$^{ts1}$, and *Df(1)J5*; *l(3)mbt*$^{ts1}$; D = genes specifically affected in *Df(1)J5*; *l(3)mbt*$^{ts1}$. Data Information: Venn Diagrams were performed using BioVenn online tool, and Gene Ontology terms were analysed using DAVID 2021 online tool. Statistical analysis performed by DAVID was Fisher's Exact test.

