## [Peer Review File · EMBO Reports]

TrxT and dhd are dispensable for Drosophila brain development but essential for l(3)mbt brain tumour growth

Cristina Molnar, Jan Heinen, Jose Reina, Salud Llamazares, Emilio Palumbo, Giulia Pollarolo, and Cayetano Gonzalez

Corresponding author(s): Cayetano Gonzalez (gonzalez@irbbarcelona.org)

Review Timeline:

Transfer Date:	4th Mar 24
Editorial Decision:	26th Mar 24
Revision Received:	8th Apr 24
Editorial Decision:	22nd Apr 24
Revision Received:	25th Apr 24
Accepted:	26th Apr 24

Editor: Achim Breiling

Transaction Report: This manuscript was transferred to EMBO reports following peer review at Review Commons.

Review
COMMONS

Review #1

1. Evidence, reproducibility and clarity:

Evidence, reproducibility and clarity (Required)

Two cancer-germline (CG) genes encoding the *Drosophila* thioredoxins Deadhead (Dhd) and Thioredoxin-T (TrxT) are located head-to-head in the X chromosome. Cristina Molnar and coworkers investigate the effects of Dhd and TrxT in brain tumours of either sex caused by mutations in l(3)malignant brain tumour (l(3)mbt). Using CRISPR/Cas9-mediated knock-out alleles and RNA-seq, they demonstrate that, although both TrxT and Dhd are not required for normal brain development, they have a significant but partial effect on l(3)mbt brain tumour development, that is stronger in male than in female larval brains. However, allograft experiments show that only TrxT plays a significant role in long-term, sustained tumour growth. TrxT and dhd play a synergistic contribution role in development of mbt tumours and in the emergence of l(3)mbt tumour-linked transcriptomic signatures.

****Major comments:****

Most of the work in this paper is well conducted and the key conclusions are convincing. I think that the number of the replicates/animals for the experiments described in Figures 1 and 2 should be reported either in the figure legends or in the methods (statistical analysis).

A relevant part of the discussion repeats what the authors have already said in the results.

I would recommend to reorganize this section, emphasizing the importance of these results in the context of human brain tumors.

2. Significance:

Significance (Required)

This work provides the first instance of an X-linked, head-to-head cancer-germline gene pair in *Drosophila* showing that these genes are dispensable for somatic cell development but have a crucial role to prevent malignant growth. Importantly, in humans, cancer germline genes and cancer testis (CT) genes have been involved in a wide range of cancers and about half of CT genes are located on the X chromosome. Thus, findings in this paper would be of interest to a broad audience that includes all the scientists studying the molecular mechanisms leading to cancer development.

The following keywords describe my expertise: Drosophila genetics, cell division, cancer genetics. I have less expertise to evaluate transcriptomics.

3. How much time do you estimate the authors will need to complete the suggested revisions:

Estimated time to Complete Revisions (Required)

(Decision Recommendation)

Less than 1 month

Yes

Review #2

1. Evidence, reproducibility and clarity:

Evidence, reproducibility and clarity (Required)

The manuscript by Molnar and colleagues is a follow-up from the authors' previously published work (Molnar et al 2019), where they demonstrated differential sex-linked gene expression in Drosophila l(3)mbt brain tumours and identified the thioredoxin proteins TrxT and Dhd in the mbt tumour signature (MBTS).

In the current manuscript, the authors generated genetic mutants for both genes using CRISPR TrxTKO and dhdKO and a deficiency that spans both genes (Df(1)J5) to understand the role played by these thioredoxins in normal larval brain development and l(3)mbt tumour development. Both genes were found to be largely dispensable for

normal larval brain development, however the authors uncovered a role in *l(3)mbt* tumour growth and development. Interestingly, although both *TrxT* and *Dhd* were required for *l(3)mbt* tumour development, they appeared to have distinct roles in long-term tumour growth, as assessed by allograft-induced lethality. Furthermore, the authors also investigated the link between the sexually dimorphic signatures of *l(3)mbt* tumours and the thioredoxins and suggest that both genes are required for the differential gene expression observed between the sexes.

Some of the conclusions are fairly well supported by the data presented. However, there are some aspects that are potentially not fully explored and that would provide more weight to the claims made by the authors. Some of these can be addressed by clarifications made in the text and/or figures and others would benefit from additional immunofluorescence staining experiments.

Specific comments are provided below.

1. Figures should include information regarding the sex of the larvae, particularly as there has been a previously reported sex-linked effect in the phenotypes analysed. (e.g. in Figure 2 and Figure S1, where Indication of the sex of the animals should be provided in the figure and not just in the figure legend).
2. Data regarding fertility. Can this be shown in a table format? Are *dhd*KO females fully sterile? What are the fertility levels of *Df(1)J5*?
3. Are *dhd* and *TrxT* the only genes affected by *Df(1)J5*? Is there transcriptional data from *Df(1)J5* animals to suggest that nearby genes are not affected by the deficiency? Of particular interest would be to assess if *snf* is affected or not as it is a known regulator of gene expression and splicing.
4. In Figure 1C, statistical test plus indication of significance is not presented.
5. Related to Figure 1D. Additional neural markers could be assessed in *dhd*KO and *TrxTKO* flies. Whilst the gross morphology of the brain does not seem to be affected, there is a possibility that cell specification is affected. Specific markers for the NE, MED and CB could be used to assess this in more detail, particularly as the DE-cad images shown for *dhd*KO and *TrxTKO* flies seem to differ slightly from the control.
6. Related to Figure 2A, images from *TrxTKO*; *l(3)mbtts1*, *dhd*KO and *l(3)mbtts1* should be added at the very least in a supplementary figure. Additionally, data for NE/BL ratio should be provided for *dhd*KO, *TrxTKO* and *Df(1)J5* in the absence of *l(3)mbtts1* tumours. Related to Figure S1, quantification of NE/BL ratio for female

lobes should be added to the figure.

7. Related to Figure 2B and Figure S1, three rows of images are presented for each genotype. It is unclear whether these correspond to brain lobes from different larvae or different confocal planes from the same animal. This should be clarified in the figure and/or figure legend.

Related to this, in addition to the anti-DE-cadherin data, it would be informative to include immunofluorescence data using antibodies such as anti-Dachshund (lamina), anti-Elav (medulla cortex) and anti-Prospero (central brain and boundary between central brain and medulla cortex) (as assessed in e.g. Zhou and Luo, J Neurosci 2013) in the mbt tumour situation to accurately describe regions disrupted by the tumours.

8. Authors should clarify how the NE was defined when mbt tumours are generated, as it is severely affected. From the images provided, it is unclear which region corresponds to NE or how the NE/BL ratio was measured. It would be helpful to outline these regions in the images or, as mentioned above, use antibodies to define them.

9. Figure 2C does not have indication of statistical significance for the comparisons stated in the text. Potential explanations for the different roles of Dhd and TrxT in long-term tumour development should be explored in the discussion. Related to this, does the analysis of the RNA-seq data from TrxTKO; l(3)mbtts1 and dhdKO; l(3)mbtts1 animals reveal why they have similar effect on mbt tumour development but do not synergistically contribute to long-term growth?

10. Authors should clarify if there is any overlap between the affected M-tSDS and F-tSDS in the TrxTKO; l(3)mbtts1 and dhdKO; l(3)mbtts1 conditions. Would the limited overlap suggest that TrxT and dhd act in parallel rather than synergistically? This might also explain the differential effects on long-term tumour development. Additionally, the stronger effect observed in Df(1)J5 animals may be due to TrxT and dhd functional redundancy. Currently, there is limited evidence to suggest that TrxT and dhd act synergistically to regulate mbt tumour growth based on the presented data.

11. Authors should include a Venn diagram depicting affected genes (M-tSDS and F-tSDS) in the TrxTKO; l(3)mbtts1, dhdKO; l(3)mbtts1 and Df(1)J5; l(3)mbtts1 genotypes as this could clarify the percentage of overlap of gene signatures in these different conditions. Related to this point, authors could provide results from GO analysis to investigate whether specific functional clusters are altered in the different conditions.

12. In Figure 3E, authors should indicate more explicitly in the figure panel and/or figure legend which genes display significant differences in expression in the different samples.

13. In Figure S2C-F it is not clear if the graphs represent data from all tissues or data from male and female tissues separately, as shown in Figure 4.

14. Are TrxT and dhd also deregulated in other tumour types? Or is this specific for mbt tumours? This information could be provided to enhance the scope of the manuscript.

15. Authors conclude that TrxT and dhd cooperate in controlling gene expression between wild-type and tumour samples and that they act synergistically in the regulation of sex-linked gene expression in male tumour tissue. However, the link between the two observations (if indeed there is a link) has not been well explained. Is the effect on gene expression in tumours simply a result of the regulation of sex-linked transcription?

2. Significance:

Significance (Required)

The current manuscript provides additional information regarding the regulation of mbt tumours and establishes TrxT and Dhd as potential cancer-germline genes in *Drosophila*. This will be of interest to researchers studying basic mechanisms of tumourigenesis and could potentially lead to identification of genes that could serve as biomarkers of disease. However, for this, a more general role of TrxT and Dhd needs to be established, as well as their potential conserved role as cancer-germline genes needs to be established.

3. How much time do you estimate the authors will need to complete the suggested revisions:

Estimated time to Complete Revisions (Required)

(Decision Recommendation)

Between 1 and 3 months

Yes

Review #3

1. Evidence, reproducibility and clarity:

Evidence, reproducibility and clarity (Required)

****Summary:****

- In this manuscript, the authors address the role of the thioredoxins Dhd and TrxT in the development and growth of mbt tumors, a sexually dimorphic brain tumor that derives from the expansion of the neuroepithelium. To this end, the authors have successfully generated dhd and TrxT knock-out mutants using CRISPR-Cas9 and show that both dhd and TrxT individual knock-out partially reduces the mbt tumor-associated brain phenotype. Moreover, using Df(1)J5, a deficiency that affects both TrxT and dhd, the recovery of the phenotype is enhanced. However, although concomitant expression of dhd and TrxT is required for proper tumor development, they show that only TrxT is necessary for the growth of allografts derived from male l(3)mbt tumors. This is interesting, not only because TrxT and dhd are never co-expressed in physiological conditions, but also because this data suggests that the pathways leading to l(3)mbt tumor development are different from the ones that contribute to tumor proliferation and aggressiveness. Moreover, the authors show that TrxT and dhd contribute to the emergence of the mbt tumour signature (MBTS) and sex-dimorphic signature (SDS) of tumours by analysing transcriptomic data of TrxT KO; l(3)mbt, dhd KO; l(3)mbt and Df(1)J5; l(3)mbt. In fact, through hierarchical clustering, the authors show that male Df(1)J5; l(3)mbt brain transcriptomic profile becomes closer to wild-type brains than l(3)mbt ts1 tumors.

- This study presents novelty to the cancer research field and both the model and methodology used were appropriate. Nonetheless, this study deals with mbt tumors which are sexually dimorphic, as well as male and female germline-specific genes that in a tumor can alter male and female sex-dimorphic signatures, making this study very easy to become confusing to non-experts in the field if not written in a very clear way. Therefore, the text, especially in the results and discussion section, could be revised in general to improve the comprehension and flow of the manuscript, given that some sentences and paragraphs are hard to follow. In particular, the results section could benefit with more contextualization and a more detailed explanation of experiments. Moreover, the study is lacking some quantifications and a few additional experiments. These issues can certainly be addressed by reviewing the text as well as reorganizing and including a few quantifications and experiments as described below. I am an expert in *Drosophila* brain development and tumorigenesis.

****Comments:****

- In the first section of the results, as a first step to study the role of TrxT and dhd genes on mbt tumors the authors generate CRISPR knock outs of these genes and correctly validate them. However, afterwards, the experiment where the authors test the KO of these genes in a wild-type larva brain is not contextualized with the rest of the section. It might be best to first address the role of these genes in a tumor context and only then complement with the experiments in wild-type (in supplementary material).

- Fig 2 B - To back up the quantifications in Fig 2A the authors could include images of l(3)mbt ts1 tumors with TrxT KO and dhd KO also.

Fig 2 B and C - Indeed, the results suggest that TrxT seems to be responsible for most tumor lethality upon l(3)mbt allografts, but not dhd. This is curious since l(3)mbt; dhd KO brain tumors have the same partial phenotype as l(3)mbt; TrxT KO (fig 1A). It would be interesting to further explore these phenotypes by staining l(3)mbt; TrxT KO and l(3)mbt; dhd KO brains with, for instance, PH3 to understand if the number of dividing cells of these tumors could be different. In addition, to back up this information, the authors could look at what happens to l(3)mbt tumors with TrxT KO and dhd KO at a later stage of development (or to larva or pupa lethality if that is the case) and compare it with l(3)mbt brains.

- Fig 2 B - What happens to the medulla in a l(3)mbt brain tumor? Although the ratio of NE/BL is the same for wild-type and D(1)J5; l(3)mbt, it still seems that the medulla in D(1)J5; l(3)mbt brains is substantially bigger, although quantifications would be required. Do the authors know if the NE in D(1)J5; l(3)mbt brains is either

proliferating less or differentiating more?

- Figure S1 - Although the effects of TrxT KO and dhd KO in male mbt tumors seem to be enhanced in relation to female tumors, the authors should include some form of tumor quantification for female tumors like in Fig 2 A.

Moreover in the 2nd section of the results, relative to Fig 1S in "...Df(1)J5; l(3)mbtts1 female larvae although given the much less severe phenotype of female mbt tumours, the effect caused by Df(1)J5 is quantitatively minor." to say "quantitatively" minor, the authors should include not only quantifications, but a form of comparison between female tumors vs. male tumors.

- Fig 3D - The hierarchical clustering was done according to which parameters? A brief explanation could help a better interpretation of this results section.

- Fig 3D - It could be beneficial for the authors to include an analysis of the downregulated genes shared between TrxT KO mbt tumors and dhd KO mbt tumors, as well as the genes that are not shared (besides MBTS genes). Could be something like a Venn diagram.

- Results section 3 - "Expression of nanos is also significantly down-regulated upon TrxT loss, but remains unaffected by loss of dhd" - to corroborate the idea that TrxT and dhd work as a pair, but contribute to different functions within the tumor, it would be interesting for the authors to do an allograft experiment of dhd KO; l(3)mbt male tissue with nanos knock down in the brain, if genetically possible.

****Minor comments:****

- In the first section of the results, the authors claim that "Consistent with the reported phenotypes of Df(1)J5...", but then the study is not mentioned.

- Fig 1 B - It is a bit confusing to follow where TrxT and dhd are in the Genome browser view. I am guessing we should follow the TrxT-dhd locus from A, but the authors could make it clearer.

- In the same section, in the next sentence, the homozygous and hemizygous is a bit confusing. "...homozygous TrxTKO females, dhdKO males, and TrxTKO males", should be corrected.

- In the same section (Fig 1C): "RNA-seq data also shows that TrxT is significantly upregulated in l(3)mbtts1 males compared to females (FC=7.06; FDR=1.10E-44)

while *dhd* is not (FC=1.89; FDR=2.00E-14)." - But *dhd* is nevertheless upregulated, although less, in *l3mbt* males, right? The authors might need to rephrase.

- Fig 2 A (quantifications), should be after the confocal images (Fig 2 B).

- Fig 2 B and Fig S1 - Please include an outline of at least neuroepithelia and, if possible, Central brain or medulla so that these regions can more clearly identified. Moreover, these results will be easier to interpret if you add a male symbol in this image and a female symbol in Figure S1, otherwise, it might seem like the same figure if one does not properly read the legend.

- In results, section 2, "Consequently, in spite of the strong sex dimorphism of *mbt* tumours, the phenotype of *Df(1)J5; l(3)mbtts1* larval brains is not sexually dimorph" - to back this up, quantifications of *Df(1)J5; l(3)mbtts1* female vs male tumor size, as well as statistical analysis are needed, like previously said.

- In results section 2 - "For allografts derived from female larvae, we found that differences in lethality rate caused by *TrxTKO; l(3)mbtts1*, *dhdKO; l(3)mbtts1*, *Df(1)J5; l(3)mbtts1*, and *l(3)mbtts1* tissues (7-23%) were not significant (Figure 2C)" - there is no statistical analysis to conclude that the lethality rate is not significant, from 7% to 23% still seems like a difference.

- Last paragraph of section 2 of results - very long and confusing sentence. Please rephrase text.

- On section 3 of results: "The *vas*, *piwi* and *CG15930* transcripts are not significantly down-regulated following either *TrxT* or *dhd* depletion alone." - in Fig 3E, not only these transcripts seem to suffer a slight downregulation, but there is also no statistical analysis supporting this.

- First paragraph of section 3 results - the first sentence is written in a confusing way. Moreover, more context is needed in the sentence afterwards: "we first focused on transcripts that are up-regulated in male *mbt* tumour samples compared to male wild-type larval brains (*mMBTS*)." but using which data? The RNA seq data?

- Brief conclusion missing on the second paragraph of the last section of results.

- In the end of 3rd paragraph of last section of results: "...M-tSDS and F-tSDS genes is partially reduced in *l(3)mbtts1* brains lacking either *TrxT* or *dhd*, but it is completely suppressed upon the lack of both." - "completely" might not be a correct word to use in this case, as there is still some small differences.

- 4th paragraph of last section of results: Either mention the male results and then female (to be in order with the figure, as the female graphs come after the male graphs) or change the order in the figure. Also, this paragraph is not very clear, could benefit from a better explanation of the results and conclusions.

- Fig 4 C,D,E,F: to make it more clear, please write the name of the genotypes in question in the figure.

2. Significance:

Significance (Required)

This study presents an interesting new concept for *Drosophila* tumors, the cancer germline genes, which to my knowledge has been a poorly explored field, although it has a lot of potential. It is particularly interesting since it addresses the role of two germline specific thioredoxins, that are dispensable for somatic cells, but have a critical role in somatic mbt tumors, exploring new tumor vulnerabilities. This manuscript will benefit researchers in the field of cancer biology, in particular, to better understand cancer-testis (CT) genes and how they promote tumorigenesis, since the biological function for the most part remains unclear.

3. How much time do you estimate the authors will need to complete the suggested revisions:

Estimated time to Complete Revisions (Required)

(Decision Recommendation)

Between 1 and 3 months

No

Full Revision

Manuscript number: RC-2023-02273R

Corresponding author(s): Cayetano Gonzalez

1. General Statements [optional]

We very much appreciate the reviewers' thorough comments and are sincerely grateful for their kind remarks on the novelty and interest of our manuscript. We are confident to have addressed all the points that they have raised including new data, as well as revised figures and text.

All the revisions have been already carried out and included in the transferred manuscript.

Reviewer #1

Major comments:

> *The number of the replicates/animals for the experiments described in Figures 1 and 2 should be reported either in the figure legends or in the methods (statistical analysis).*

We have added the required numbers to the corresponding revised figures, as requested.

> *A relevant part of the discussion repeats what the authors have already said in the results. I would recommend to reorganize this section, emphasizing the importance of these results in the context of human brain tumors.*

Following our own style, we have written a very short (46 lines in length!) Discussion. We dedicate a few lines to highlighting two points: (1) the suggestion, derived from our allograft experiments, that the initial stages of tumour development and long-term tumour growth may be molecularly distinct events, and (2), the unique effect of the combined loss of *TrxT* and *dhd* on mbt tumour transcriptomics -unique because none of the suppressors of mbt reported before are as effective in erasing both the MBTS and SDS mbt signatures. Neither of these points are raised in Results. In the remaining few lines we put our results in the context of human Cancer/Testis and elaborate on the fact that the *TrxT* and *dhd* pair qualify as head-to-head, CT-X genes, like those reported in human oncology. This is as far as we are willing to go at this stage at emphasizing the importance of our results in the context of human tumours.

Full Revision

Reviewer #2

> 1. *Figures should include information regarding the sex of the larvae, particularly as there has been a previously reported sex-linked effect in the phenotypes analysed. (e.g. in Figure 2 and Figure S1, where Indication of the sex of the animals should be provided in the figure OK and not just in the figure legend).*

We fully agree. Sex must always be taken into account as a biological variable. All the experiments reported in the manuscript were carried out with sexed samples, and were annotated accordingly in the original text. In compliance with the reviewer's request we have added this information also to the revised figure.

> 2. *Data regarding fertility. Can this be shown in a table format? Are dhdKO females fully sterile? What are the fertility levels of Df(1)J5?*

Please note that we are not discovering anything here but merely corroborating what has been published before: the lack of TrxT does not affect fertility in either sex; the lack of Dhd results in female sterility (Torres-Campana et al., 2022, Tirmarche et al., 2016, Svensson et al., 2003, Pellicena-Palle et al., 1997). Adding a table would not be justified. Moreover, it would be a rather simple table: all single-pair mating tests (n=10 for each genotype) with TrxT KO and Dhd KO males, and TrxT KO females were as fertile as control flies, while all single-pair mating tests (n=10) with Dhd KO females were sterile.

> 3. *Are dhd and TrxT the only genes affected by Df(1)J5? Is there transcriptional data from Df(1)J5 animals to suggest that nearby genes are not affected by the deficiency? Of particular interest would be to assess if snf is affected or not as it is a known regulator of gene expression and splicing.*

Yes dhd and TrxT are the only genes affected by Df(1)J5. That is the case according to Flybase (citing Svensson et al., 2003, and Salz et al., 1994) and confirmed by our own RNAseq data. No other transcripts, including snf, are affected by Df(1)J5.

> 4. *In Figure 1C, statistical test plus indication of significance is not presented.*

The requested statistical test and significance data have been added as required to the revised figure and figure legend.

> 5. *Related to Figure 1D. Additional neural markers could be assessed in dhdKO and TrxTKO flies. Whilst the gross morphology of the brain does not seem to be affected, there is a possibility that cell specification is affected. Specific markers for the NE, MED and CB could be used to assess this in more detail, particularly as the DE-cad images shown for dhdKO and TrxTKO flies seem to differ slightly from the control.*

We believe that there may be a small misunderstanding here. We have made this point clear in the revised version by referring to substantial published data showing that expression of these two genes is restricted to the germline and that, female fertility aside, TrxT and dhd deficient flies' development and life span are perfectly normal. If anything, Figure 1D is redundant. However, we would rather keep it as a control that our CRISPR KO mutants behave as expected.

> 6. *Related to Figure 2A, images from TrxTKO; l(3)mbtts1, dhdKO and l(3)mbtts1 should be added at the very least in a supplementary figure. Additionally, data for NE/BL ratio should be provided for dhdKO, TrxTKO and Df(1)J5 in the*

Full Revision

absence of $l(3)mbtts1$ tumours. Related to Figure S1, quantification of NE/BL ratio for female lobes should be added to the figure.

All the requested images and data have been included in the revised version in new figures Figure S2B, Figure S2A, and Figure S1A.

> 7. Related to Figure 2B and Figure S1, three rows of images are presented for each genotype. It is unclear whether these correspond to brain lobes from different larvae or different confocal planes from the same animal. This should be clarified in the figure and/or figure legend.

This point has been clarified as requested in the revised figure legend. Each group of three rows correspond to brain lobes from different larvae of the same genotype.

> 7 cont. Related to this, in addition to the anti-DE-cadherin data, it would be informative to include immunofluorescence data using antibodies such as anti-Dachshund (lamina), anti-Elav (medulla cortex) and anti-Prospero (central brain and boundary between central brain and medulla cortex) (as assessed in e.g. Zhou and Luo, J Neurosci 2013) in the mbt tumour situation to accurately describe regions disrupted by the tumours.

There is no denying that taking advantage of the many cell-type specific markers that are readily available in *Drosophila* could be of interest. The same applies to cell cycle markers like PH3, FUCCI, and many others. However, we believe that interesting as they may be, none of these markers will give us the clue on the molecular basis of TrxT and Dhd tumour function that is, of course, the open burning question that we are trying to address now.

> 8. Authors should clarify how the NE was defined when mbt tumours are generated, as it is severely affected. From the images provided, it is unclear which region corresponds to NE or how the NE/BL ratio was measured. It would be helpful to outline these regions in the images or, as mentioned above, use antibodies to define them.

The figure has been modified to include the requested outlines defining the NE that indeed is correspond to the channel showing DE-Cadh staining.

> 9. Figure 2C does not have indication of statistical significance for the comparisons stated in the text. Potential explanations for the different roles of Dhd and TrxT in long-term tumour development should be explored in the discussion.

The requested statistical significance data for these comparisons were stated in the second last paragraph of that section. To make these data more prominent we have also added this information to revised Figure 2C.

> 9 cont. Related to this, does the analysis of the RNA-seq data from TrxTKO; $l(3)mbtts1$ and dhdKO; $l(3)mbtts1$ animals reveal why they have similar effect on mbt tumour development but do not synergistically contribute to long-term growth?

Unfortunately our analysis of the RNA-seq data from TrxTKO; $l(3)mbtts1$ and dhdKO; $l(3)mbtts1$ animals does not give us any clue that could help us understand why they have similar effect on mbt tumour development, but not in long-term growth (allografts). To further explore this point, we have added new Figure S3 that includes a Venn diagram showing the overlap between the affected mMBTS genes in TrxTKO; $l(3)mbtts1$ and dhdKO; $l(3)mbtts1$, together with the lists of enriched GOs among overlapping and non-overlapping genes. GO differences are

Full Revision

tantalising, indeed, However, they do not immediately suggest any direct explanation for the different roles of Dhd and TrxT in long-term tumour development.

> 10. Authors should clarify if there is any overlap between the affected M-tSDS and F-tSDS in the TrxTKO; *l(3)mbtts1* and *dhdKO*; *l(3)mbtts1* conditions. Would the limited overlap suggest that TrxT and *dhd* act in parallel rather than synergistically? This might also explain the differential effects on long-term tumour development. Additionally, the stronger effect observed in *Df(1)J5* animals may be due to TrxT and *dhd* functional redundancy. Currently, there is limited evidence to suggest that TrxT and *dhd* act synergistically to regulate mbt tumour growth based on the presented data.

See below.

> 11. Authors should include a Venn diagram depicting affected genes (M-tSDS and F-tSDS) in the TrxTKO; *l(3)mbtts1*, *dhdKO*; *l(3)mbtts1* and *Df(1)J5*; *l(3)mbtts1* genotypes as this could clarify the percentage of overlap of gene signatures in these different conditions. Related to this point, authors could provide results from GO analysis to investigate whether specific functional clusters are altered in the different conditions.

We have taken the liberty of fusing points 10 and 11 that are conceptually similar. The requested Venn diagrams showing the overlap between the affected M-tSDS and F-tSDS genes in the TrxTKO; *l(3)mbtts1*, *dhdKO*; *l(3)mbtts1*, and *Df(1)J5*; *l(3)mbtts1* conditions, and GO analysis are now shown in new Figure S5. Unfortunately, these new data do not suggest any obvious explanation for the differential effects of these two genes, nor do they allow us to derive any further conclusions regarding the nature of the pathways through which TrxT and *dhd* cooperate to sustain mbt tumour growth. However, our analyses demonstrate that efficient suppression of mbt phenotypic traits (in larval brains) and transcriptome requires the combined elimination of both germline thioredoxins, while the effect of individual removal of either of them is only partial. These data demonstrate the synergistic nature of TrxT and *dhd* function in mbt tumour growth.

> 12. In Figure 3E, authors should indicate more explicitly in the figure panel and/or figure legend which genes display significant differences in expression in the different samples.

We apologise for not having made this point clear in the original version: All (21) genes shown in this Table are significantly downregulated in *DfJ5;ts1* vs *ts1*. From these, *nanos* and *Ocho* are also significantly downregulated in *TrxTKO;ts1* vs *ts1*, and *Ocho*, *HP1D3csd*, *hlk*, *fj*, *Lcp9*, *CG43394*, and *CG14968* are significantly downregulated in *dhdKO;ts1* vs *ts1*. These data have been included in the revised figure legend. Data on all other comparisons are included in Table S1.

> 13. In Figure S2C-F it is not clear if the graphs represent data from all tissues or data from male and female tissues separately, as shown in Figure 4.

Apologies for the confusion. All samples were from male tissues as indicated in the original figure legend. To make it more clear, we have labelled all four panels in the revised figure.

> 14. Are TrxT and *dhd* also deregulated in other tumour types? Or is this specific for mbt tumours? This information could be provided to enhance the scope of the manuscript.

Full Revision

Thank you for raising this point. TrxT and dhd are not dysregulated in the other tumour types that were analysed in Janic et al., 2010 (i.e pros, mira, brat, lgl and pins).

> 15. *Authors conclude that TrxT and dhd cooperate in controlling gene expression between wild-type and tumour samples and that they act synergistically in the regulation of sex-linked gene expression in male tumour tissue. However, the link between the two observations (if indeed there is a link) has not been well explained. Is the effect on gene expression in tumours simply a result of the regulation of sex-linked transcription?*

Our data show that TrxT and dhd synergistically contribute to the emergence of both the MBTS (i.e tumour versus wild type) and SDS (i.e. male tumour versus female tumour). The only certainty at this time regarding the interconnection between both signatures is that they overlap, but only partially, which answers one the questions raised by the reviewer: the effect on gene expression in tumours is not simply a result of the regulation of sex-linked transcription. Beyond that, the link (if indeed there is a link) between these two signatures has not been investigated. The lack of insight on this issue is not surprising taking into account that, in contrast to classical tumour signatures (tumour versus healthy tissue), the concept of sex-linked tumour signatures is relatively new and only a handful of such signatures have been published. Moreover, the vast majority of classical tumour signatures have not been worked out in a sex-dependent manner.

Reviewer #3

Comments:

> - *In the first section of the results, as a first step to study the role of TrxT and dhd genes on mbt tumors the authors generate CRISPR knock outs of these genes and correctly validate them. However, afterwards, the experiment where the authors test the KO of these genes in a wild-type larva brain is not contextualized with the rest of the section. It might be best to first address the role of these genes in a tumor context and only then complement with the experiments in wild-type (in supplementary material).*

We do appreciate the reviewer's view, but respectfully disagree. In our opinion, the manuscript flows better by presenting the tools that we have generated in Figure 1, By corroborating published data showing that these two germline genes do not affect soma development (Torres-Campana et al., 2022, Tirmarche et al., 2016, Svensson et al., 2003, Pellicena-Palle et al., 1997) this first figure not only validates our CRISPR KO mutants, but also sets the stage to highlight their significant effect on a somatic tumour like mbt.

> - *Fig 2 B - To back up the quantifications in Fig 2A the authors could include images of l(3)mbt ts1 tumors with TrxT KO and dhd KO also.*

The requested images are shown in new figure Figure S2B.

> *Fig 2 B and C - Indeed, the results suggest that TrxT seems to be responsible for most tumor lethality upon l(3)mbt allografts, but not dhd. This is curious since l(3)mbt; dhd KO brain tumors have the same partial phenotype as l(3)mbt; TrxT KO (fig 1A). It would be interesting to further explore these phenotypes by staining l(3)mbt; TrxT KO and l(3)mbt; dhd KO brains with, for instance, PH3 to understand if the number of dividing cells of these tumors could be different.*

Full Revision

*In addition, to back up this information, the authors could look at what happens to *l(3)mbt* tumors with *TrxT* KO and *dhd* KO at a later stage of development (or to larva or pupa lethality if that is the case) and compare it with *l(3)mbt* brains.*

We did explore the possibility of looking at later stages. Unfortunately, the onset of the lethality phase compounded by major tissue reshaping from larval to adult brain make these stages unsuitable to reach any meaningful conclusion. With regards to staining for PH3, we think that like FUCCI and a long list of other useful labels that could be explored, it is potentially interesting, but hardly likely to give us the clue on the molecular basis of *TrxT* and *Dhd* tumour function, that is of course the one important question that we are addressing now.

*> - Fig 2 B - What happens to the medulla in a *l(3)mbt* brain tumor? Although the ratio of NE/BL is the same for wild-type and *D(1)J5; l(3)mbt*, it still seems that the medulla in *D(1)J5; l(3)mbt* brains is substantially bigger, although quantifications would be required. Do the authors know if the NE in *D(1)J5; l(3)mbt* brains is either proliferating less or differentiating more?*

There are no significant differences in medulla/BL nor in CB/BL ratios. The corresponding quantifications have been added to the revised version. As for the question on proliferation versus differentiation, the simple answer is that we do not know.

*- Figure S1 - Although the effects of *TrxT* KO and *dhd* KO in male *mbt* tumors seem to be enhanced in relation to female tumors, the authors should include some form of tumor quantification for female tumors like in Fig 2 A.*

We have carried out the requested quantifications and added the results in a new panel in revised Figure S1A.

*> Moreover in the 2nd section of the results, relative to Fig 1S in "...*Df(1)J5; l(3)mbtts1* female larvae although given the much less severe phenotype of female *mbt* tumours, the effect caused by *Df(1)J5* is quantitatively minor." to say "quantitatively" minor, the authors should include not only quantifications, but a form of comparison between female tumors vs. male tumors.*

The requested quantification was published in Molnar et al., 2019. However, we agree on the convenience of doing it again with our new samples. The new data, that confirm published results, are now shown as a new panel in revised Figure S1C.

> - Fig 3D - The hierarchical clustering was done according to which parameters? A brief explanation could help a better interpretation of this results section.

The requested information has been added to the Methods section. Hierarchical clustering was done using the function `heatmap.2` in R to generates a plot in which samples (columns) are clustered (dendrogram); genes (rows) are scaled by "rows"; distance = Euclidean; and `hclust` method = complete linkage. Expression levels are reported as Row Z-score.

*> - Fig 3D - It could be beneficial for the authors to include an analysis of the downregulated genes shared between *TrxT* KO *mbt* tumors and *dhd* KO *mbt* tumors, as well as the genes that are not shared (besides MBTS genes). Could be something like a Venn diagram.*

Full Revision

Thanks for pointing this out. New Figure S3 shows the requested Venn diagram, as well as the list of enriched GOs for each group. There are no enriched GOs in the list of overlapping genes. TrxTKO; l(3)mbtts1-specific genes are enriched for GOs related to game generation, sexual reproduction, germ cell development and similar GOs. dhdKO; l(3)mbtts1-specific genes are enriched for GOs related to chitin, molting and cuticle development. Tantalising as they are, these observations do not immediately suggest any direct explanation for the different roles of Dhd and TrxT in long-term tumour development. We are happy to add this supplemental information, but we do not deem it worth of any further discussion at this point.

> - *Results section 3 - "Expression of nanos is also significantly down-regulated upon TrxT loss, but remains unaffected by loss of dhd" - to corroborate the idea that TrxT and dhd work as a pair, but contribute to different functions within the tumor, it would be interesting for the authors to do an allograft experiment of dhd KO; l(3)mbt male tissue with nanos knock down in the brain, if genetically possible.*

The suggested experiment is published. The gene in question (nanos) is a suppressor of mbt tumour growth: In a nanos knock down background, l(3)mbt allografts do not grow (Janic 2010).

Minor comments:

> - *In the first section of the results, the authors claim that "Consistent with the reported phenotypes of Df(1)J5...", but then the study is not mentioned.*

The corresponding references (Salz et al., 1994; Svensson et al., 2003; Tirmarche et al., 2016) have been added.

> - *Fig 1 B - It is a bit confusing to follow where TrxT and dhd are in the Genome browser view. I am guessing we should follow the TrxT-dhd locus from A, but the authors could make it clearer.*

Figure 1 has been changed to make this point more clear.

> - *In the same section, in the next sentence, the homozygous and hemizygous is a bit confusing. "...homozygous TrxTKO females, dhdKO males, and TrxTKO males", should be corrected.*

We appreciate the suggestion, but would rather stick to classical terminology and refer to KO/KO females as homozygous and to KO/Y males as hemizygous.

- *In the same section (Fig 1C): "RNA-seq data also shows that TrxT is significantly upregulated in l(3)mbtts1 males compared to females (FC=7.06; FDR=1.10E-44) while dhd is not (FC=1.89; FDR=2.00E-14)." - But dhd is nevertheless upregulated, although less, in l3mbt males, right? The authors might need to rephrase.*

We refer to comparing males versus females, not wild type versus tumours. The text has been rephrased in the revised version to make this point clear.

> - *Fig 2 A (quantifications), should be after the confocal images (Fig 2 B).*

We respectfully disagree on this minor point. We initially organised this figure in the order recommended by the reviewer, but we eventually found it easier to write the article using the order shown in the submitted figure. We would rather stick to this version.

Full Revision

> - Fig 2 B and Fig S1 - Please include an outline of at least neuroepithelia and, if possible, Central brain or medulla so that these regions can more clearly identified. Moreover, these results will be easier to interpret if you add a male symbol in this image and a female symbol in Figure S1, otherwise, it might seem like the same figure

Outlines and symbols have been added to the revised figure, as required.

> - In results, section 2, "Consequently, in spite of the strong sex dimorphism of mbt tumours, the phenotype of *Df(1)J5; l(3)mbtts1* larval brains is not sexually dimorph" - to back this up, quantifications of *Df(1)J5; l(3)mbtts1* female vs male tumor size, as well as statistical analysis are needed, like previously said.

The requested the new data is now shown in revised Figure S1C.

> - In results section 2 - "For allografts derived from, female larvae, we found that differences in lethality rate caused by *TrxTKO; l(3)mbtts1*, *dhdKO; l(3)mbtts1*, *Df(1)J5; l(3)mbtts1*, and *l(3)mbtts1* tissues (7-23%) were not significant (Figure 2C)" - there is no statistical analysis to conclude that the lethality rate is not significant, from 7% to 23% still seems like a difference.

Thanks for pointing this out. We did of course generate the requested statistical analysis data, but failed to include it in the manuscript. Chi-square statistical test gives a p value=0.2346. These data have been added to the revised version.

> - Last paragraph of section 2 of results - very long and confusing sentence. Please rephrase text.

We have rephrased this sentence to make it shorter and clearer.

> - On section 3 of results: "The *vas*, *piwi* and *CG15930* transcripts are not significantly down-regulated following either *TrxT* or *dhd* depletion alone." - in Fig 3E, not only these transcripts seem to suffer a slight downregulation, but there is also no statistical analysis supporting this.

There seems to be a misunderstanding here. The requested statistical data for each gene were shown in Table S1

> - First paragraph of section 3 results - the first sentence is written in a confusing way. Moreover, more context is needed in the sentence afterwards: "we first focused on transcripts that are up-regulated in male mbt tumour samples compared to male wild-type larval brains (*mMBTS*)." but using which data? The RNA seq data?

Agreed; this paragraph has been amended in the revised version.

- Brief conclusion missing on the second paragraph of the last section of results.

As far as the results presented in this paragraph are concerned, we can only mention the two potentially interesting observations, which were pointed out in the original version: (i) the suggestion that *nanos* upregulation could be critical for sustained mbt tumour growth upon allograft, and (ii) the fact that three genes (*vas*, *piwi* and *CG15930*), also known to be required for mbt tumour growth, are downregulated in *Df(1)J5; l(3)mbtts1*, but remain unaffected following either *TrxT* or *dhd* depletion alone. We are unable to derive any other conclusion from these observations.

Full Revision

> - In the end of 3rd paragraph of last section of results: "...M-tSDS and F-tSDS genes is partially reduced in l(3)mbtts1 brains lacking either TrxT or dhd, but it is completely suppressed upon the lack of both." - "completely" might not be a correct word to use in this case, as there is still some small differences

As requested, we have changed "completely" for "strongly".

> - 4th paragraph of last section of results: Either mention the male results and then female (to be in order with the figure, as the female graphs come after the male graphs) or change the order in the figure. Also, this paragraph is not very clear, could benefit from a better explanation of the results and conclusions.

Point taken. Figure 4 has been changed and female graphs come before male graphs. The paragraph is clearer now. The conclusion from this paragraph is included in the final paragraph of this section.

> - Fig 4 C,D,E,F: to make it more clear, please write the name of the genotypes in question in the figure.

At the reviewer's request, the genotypes in question are now written in each panel. Please note that we did not do so before because all four panels correspond to the same genotype: Df(J5); l(3)mbtts1 vs l(3)mbtts1, as we mentioned in the original figure legend.

Dear Prof. Gonzalez,

Thank you for the transfer of your revised manuscript from Review Commons to our editorial offices. I have now received the reports from the three referees that were asked to re-evaluate your study, you will find below. As you will see, the referees fully support publication of your study in EMBO reports. Nevertheless, referees #1 and #2 have some remaining concerns and suggestions to improve the manuscript I ask you to address in a final revised manuscript.

Moreover, the manuscript now also needs formatting according to our journal style. Please carefully review the instructions that follow below.

When submitting your final revised manuscript, we will require:

1) a .docx formatted version of the final manuscript text (including legends for main figures, EV figures and tables), but without the figures included. Figure legends should be compiled at the end of the manuscript text.

2) individual production quality figure files as .eps, .tif, .jpg (one file per figure), of main figures and EV figures. Please upload these as separate, individual files upon re-submission.

3) a complete author checklist, which you can download from our author guidelines (<https://www.embopress.org/page/journal/14693178/authorguide>). Please insert page numbers in the checklist to indicate where the requested information can be found in the manuscript. The completed author checklist will also be part of the RPF.

4) that primary datasets produced in this study (e.g. RNA-seq, ChIP-seq, structural and array data) are deposited in an appropriate public database.

The accession numbers and database should be listed in a formal "Data Availability" section (placed after Materials & Methods) that follows the model below. This is now mandatory (like the COI statement). Please note that the Data Availability Section is restricted to new primary data that are part of this study. This section is mandatory. As indicated above, if no primary datasets have been deposited, please state this in this section

Data availability

- RNA-Seq data: Gene Expression Omnibus GSE46843 (<https://www.ncbi.nlm.nih.gov/geo/query/acc.cgi?acc=GSE46843>)

- [data type]: [name of the resource] [accession number/identifier/doi] ([URL or identifiers.org/DATABASE:ACCESSION])

5) We now request the publication of original source data with the aim of making primary data more accessible and transparent to the reader. Our source data coordinator has already contact you to discuss which figure panels we would need source data for. I attach again the source data checklist and a FAQ with instructions.

6) Our journal encourages inclusion of *data citations in the reference list* to directly cite datasets that were re-used and obtained from public databases. Data citations in the article text are distinct from normal bibliographical citations and should directly link to the database records from which the data can be accessed. In the main text, data citations are formatted as follows: "Data ref: Smith et al, 2001" or "Data ref: NCBI Sequence Read Archive PRJNA342805, 2017". In the Reference list, data citations must be labeled with "[DATASET]". A data reference must provide the database name, accession number/identifiers and a resolvable link to the landing page from which the data can be accessed at the end of the reference. Further instructions are available at: <http://www.embopress.org/page/journal/14693178/authorguide#referencesformat>

7) Regarding data quantification and statistics, please make sure that the number "n" for how many independent experiments were performed, their nature (biological versus technical replicates), the bars and error bars (e.g. SEM, SD) and the test used to calculate p-values is indicated in the respective figure legends (also for potential EV and Appendix figures). Please also check that all the p-values are explained in the legend, and that these fit to those shown in the figure. Please provide statistical testing where applicable. Please avoid the phrase 'independent experiment', but clearly state if these were biological or technical replicates. Please also indicate (e.g. with n.s.) if testing was performed, but the differences are not significant. In case n=2, please show the data as separate datapoints without error bars and statistics. See also: <http://www.embopress.org/page/journal/14693178/authorguide#statisticalanalysis>

Please add to each legend (main and EV figures) a 'Data Information' section explaining the statistics used or providing information regarding replicates and scales.

8) Please add scale bars of similar style and thickness to microscopic images, using clearly visible black or white bars (depending on the background). Please place these in the lower right corner of the images themselves. Please do not write on or near the bars in the image but define the size in the respective figure legend.

9) Please also note our reference format:

10) We updated our journal's competing interests policy in January 2022 and request authors to consider both actual and perceived competing interests. Please review the policy <https://www.embopress.org/competing-interests> and update your competing interests if necessary. Please name this section 'Disclosure and Competing Interests Statement' and put it after the Acknowledgements section.

11) We now use CRediT to specify the contributions of each author in the journal submission system. CRediT replaces the author contribution section. Please use the free text box to provide more detailed descriptions and do NOT add an author contributions section to the manuscript text file. See also guide to authors:

<https://www.embopress.org/page/journal/14693178/authorguide#authorshippinguidelines>

12) Please add up to 5 keywords to the manuscript, move the funding information to the acknowledgements, and order the manuscript sections like this, using these names:

Title page - Abstract - Keywords - Introduction - Results - Discussion - Methods - Data availability section - Acknowledgements - Disclosure and Competing Interests Statement - References - Figure legends - Expanded View Figure legends - Tables

Please also add titles to the EV figures.

13) Table S1 is a dataset. Please upload this dataset as original excel file with a legend and a title on the first TAB. Please name this file Dataset EV1 and change the callouts accordingly.

14) Please enter all the funding information also into our submission system during resubmission and make sure this is complete and similar to the one mentioned in the acknowledgements section of the manuscript text file.

15) Please provide a final abstract with not more than 175 words.

16) Please provide a final title with not more than 100 characters (including spaces).

17) We encourage you to use 'Structured Methods', our new Materials and Methods format. According to this format, the Methods section should include a Reagents and Tools Table (listing key reagents, experimental models, software, and relevant equipment and including their sources and relevant identifiers), uploaded as separate file, followed by a Methods and Protocols section in which we encourage the authors to describe their methods using a step-by-step protocol format with bullet points, to facilitate the adoption of the methodologies across labs. More information on how to adhere to this format as well as downloadable templates (.doc or .xls) for the Reagents and Tools Table can be found in our author guidelines (section 'Structured Methods'):

In addition, I would need from you:

- a short, two-sentence summary of the manuscript (not more than 35 words).
- three to four short (!) one sentence bullet points highlighting the key findings of your study.
- a schematic summary figure (synopsis image) in jpeg or tiff format with the exact width of 550 pixels and a height of not more than 400 pixels that can be used as a visual synopsis on our website.

I look forward to seeing a revised version of your manuscript when it is ready. Please let me know if you have questions or comments regarding the revision.

Best,

Referee #1:

The revisions carried out by the authors addressed most of the issues that were raised. The missing quantifications and statistical analysis were added to the manuscript and some figures and text was changed to improve the comprehension of the study.

After addressing the issues raised by the reviewers, the authors made the necessary changes at the level of the text that allowed a better flow and understanding of the study. Furthermore, there were some issues on figure clarity (such as Figure 1A) and figure order (such as in Figure 4) that were resolved by the authors and have improved figure comprehension. Overall, the manuscript has clarity and it is well supported by the presented data. However, there was only one minor issue that can very easily be solved:

In the revision point: "In results, section 2, "Consequently, in spite of the strong sex dimorphism of mbt tumours, the phenotype of Df(1)J5; l(3)mbtts1 larval brains is not sexually dimorph" - to back this up, quantifications of Df(1)J5; l(3)mbtts1 female vs male tumor size, as well as statistical analysis are needed, like previously said. "

- The authors have indeed included the requested data in the figure (Fig S1 C). However, according to this data, there is a significant difference between Df(1)J5; l(3)mbtts1 larval brains in female vs male, meaning that there is still some degree of sexual dimorphism, although to a lesser extent. The authors could try to make the text in accordance with the figure. Or was this data previously published already? If so, the authors can just cite it.

As this is just a minor problem that the authors can easily change and does not compromise the scientific integrity of the manuscript, I recommend this manuscript for publication without further need to send it again for revision.

Referee #2:

The authors have addressed the vast majority of the points raised by the reviewers either in the rebuttal letter, in the main text of the manuscript or by conducting further statistical analysis or providing additional data in Supplementary Figures. The current version of the manuscript is now clearer.

Minor points:

Supplementary Figures do not have titles and this needs to be corrected for the final version.

Supplementary Figure 1 has the word 'Text' written between panels B and C and this is likely to be an error.

Referee #3:

The revised version of the manuscript has fully addressed my previous requests. I think it can be published in Embo Reports.

All editorial and formatting issues were resolved by the authors.

Dear Prof. Gonzalez,

Thank you for the submission of your revised manuscript to our editorial offices. Before we can proceed with formal acceptance, I have these few editorial requests I ask you to address in a final revised manuscript:

- Please change the title (as agreed on) also in the manuscript text file:

TrxT and dhd are dispensable for Drosophila brain development but essential for l(3)mbt brain tumour growth

- We plan to publish your manuscript in the Report format, as there are not more than 5 main and EV figures. For a Scientific Report we require that results and discussion sections are combined in a single chapter called "Results & Discussion". Please do this for your manuscript. For more details, please refer to our guide to authors:

<http://www.embopress.org/page/journal/14693178/authorguide#researcharticleguide>

- Please make sure that the number "n" for how many independent experiments were performed, their nature (biological versus technical replicates), the bars and error bars (e.g. SEM, SD) and the test used to calculate p-values is indicated in the respective figure legends (also for potential EV figures and all those in the final Appendix). Please also check that all the p-values are explained in the legend, and that these fit to those shown in the figure. Please provide statistical testing where applicable. Please avoid the phrase 'independent experiment', but clearly state if these were biological or technical replicates. Please also indicate (e.g. with n.s.) if testing was performed, but the differences are not significant. In case n=2, please show the data as separate datapoints without error bars and statistics. See also:

<http://www.embopress.org/page/journal/14693178/authorguide#statisticalanalysis>

If n<5, please show single datapoints for diagrams. Moreover:

- Please indicate the statistical test used for data analysis in the legends of figures 3a-c; 4c-f; EV 3; EV 4c-f; EV 5a-b.

- Please note that the box plots need to be defined in terms of minima, maxima, centre, bounds of box and whiskers, and percentile in the legends of figure 2a; EV 1a, c; EV 2a.

- Please remove the legend for Dataset EV1 from the manuscript text file.

- Please add scale bars of similar style and thickness to all microscopic images, using clearly visible black or white bars (depending on the background). Please place these in the lower right corner of the images themselves. Please do not write on or near the bars in the image but define the size in the respective figure legend. Presently, the scale bars are rather thin. Could this be improved?

- Thanks for uploading the source data checklist. However, it seems no source data was uploaded. Please upload this as indicated in the checklist, one folder per figure.

- In the data availability section (DAS) you state that data related to this manuscript have been deposited (GEO Series accession number GSE263157). What data is that? The RNA-seq raw data? Please specify the nature of the deposited data in the DAS and also rectify the typographical error in the GEO accession number as GSE263157.

In addition, I would need from you:

- a short, two-sentence summary of the manuscript (not more than 35 words).

- two to four short (!) bullet points highlighting the key findings of your study (two lines each).

- a schematic summary figure as separate file that provides a sketch of the major findings (not a data image) in jpeg or tiff format (with the exact width of 550 pixels and a height of not more than 400 pixels) that can be used as a visual synopsis on our website.

Achim Breiling
SeniorEditor
EMBO Reports

Rev_Com_number: RC-2023-02273

New_manu_number: EMBOR-2024-59127V2

Corr_author: Gonzalez

Title: TrxT and dhd are dispensable for Drosophila brain development but essential for l(3)mbt brain tumour growth

All editorial and formatting issues were resolved by the authors.

Prof. Cayetano Gonzalez
IRB-Barcelona
Cell and Dev. Biol
Baldiri Reixac 10-12
Barcelona 08028
Spain

Dear Prof. Gonzalez,

I am very pleased to accept your manuscript for publication in the next available issue of EMBO reports. Thank you for your contribution to our journal.

Please make sure the GEO dataset will be public latest when the paper is published online.

Yours sincerely,

Achim Breiling
Sennior Editor
EMBO Reports

Rev_Com_number: RC-2023-02273

New_manu_number: EMBOR-2024-59127V3

Corr_author: Gonzalez

Title: TrxT and dhd are dispensable for Drosophila brain development but essential for I(3)mbt brain tumour growth